# Comorbidity Pattern Analysis for Predicting Amyotrophic Lateral Sclerosis

Chia-Hui Huang [1], Bak-Sau Yip [2], David Taniar [3], Chi-Shin Hwang [4] and Tun-Wen Pai [5,*]

[1] Department of Computer Science and Engineering, National Taiwan Ocean University, Keelung 20224, Taiwan; 10757030@email.ntou.edu.tw

[2] Department of Neurology, National Taiwan University Hospital Hsin Chu Branch, Hsin Chu 30059, Taiwan; neuro@hch.gov.tw

[3] Department of Software Systems and Cybersecurity, Monash University, Victoria 3800, Australia; David.Taniar@monash.edu

[4] Department of Neurology, ShuTien Hospital, Taipei 10662, Taiwan; DAH88@tpech.gov.tw

[5] Department of Computer Science and Information Engineering, National Taipei University of Technology, Taipei 10608, Taiwan

* Correspondence: twp@csie.ntut.edu.tw; Tel.: +886-2-27712171 (ext. 4222)

**Abstract:** Electronic Medical Records (EMRs) can be used to create alerts for clinicians to identify patients at risk and to provide useful information for clinical decision-making support. In this study, we proposed a novel approach for predicting Amyotrophic Lateral Sclerosis (ALS) based on comorbidities and associated indicators using EMRs. The medical histories of ALS patients were analyzed and compared with those of subjects without ALS, and the associated comorbidities were selected as features for constructing the machine learning and prediction model. We proposed a novel weighted Jaccard index (WJI) that incorporates four different machine learning techniques to construct prediction systems. Alternative prediction models were constructed based on two different levels of comorbidity: single disease codes and clustered disease codes. With an accuracy of 83.7%, sensitivity of 78.8%, specificity of 85.7%, and area under the receiver operating characteristic curve (AUC) value of 0.907 for the single disease code level, the proposed WJI outperformed the traditional Jaccard index (JI) and scoring methods. Incorporating the proposed WJI into EMRs enabled the construction of a prediction system for analyzing the risk of suffering a specific disease based on comorbidity combinatorial patterns, which could provide a fast, low-cost, and noninvasive evaluation approach for early diagnosis of a specific disease.

**Keywords:** Electronic Medical Record (EMR); disease prediction; Amyotrophic Lateral Sclerosis (ALS); weighted Jaccard index (WJI)

## 1. Introduction

Amyotrophic Lateral Sclerosis (ALS) is a multi-syndrome and fatal neurodegenerative disorder that results in progressive loss of bulbar and limb function [1]. The prevalence of ALS is uniformly distributed across most countries [2]. According to the ALS Association, the incidence rate of ALS is approximately between 1.8 and 2 per 100,000 person-years. Death from ALS occurs from respiratory failure, generally within 2–3 years from the onset of the bulbar symptom and 3–5 years after the limb onset [3]. For the majority of patients with ALS, it usually takes 9 to 15 months from symptom onset to definitive diagnosis by two or three specialists [1]. The diagnostic processes for ALS are very complicated compared to other diseases. Several steps are required to ensure that all medical evaluation items are completely performed. These processes include neurological examination and a series of diagnostic tests such as electromyography (EMG), magnetic resonance imaging (MRI), blood and urine tests, etc. However, there is yet no definitive diagnostic testing standard for ALS. Different medical evaluations are always performed after disease progression in different combinations of suggestive clinical signs of pathologies. Currently, the etiology

of sporadic ALS is unclear, and there is no cure for ALS. Treatment is mainly aimed at delaying the progression of the disease and not at eliminating the symptoms [4,5]. Hence, if at-risk subjects of ALS can be identified during the early stages, they can be advised to focus on strengthening their immune system and on improving the environmental conditions in which they live. Early prognosis may slow the progressive deterioration of ALS symptoms. Since there is a near two-year lead-time interval prior to definitive diagnosis of ALS, the symptoms and comorbidities could provide important clues for ALS prediction. Hence, we consider Electronic Medical Records (EMRs) as useful for discriminating ALS and non-ALS patients in the early stages, and the developed classifier could serve as an alternative diagnostic approach or an early warning signal for doctors and ALS patients.

This study was aimed at identifying high-risk Amyotrophic Lateral Sclerosis (ALS) subjects in the early stage using a novel detection mechanism solely based on EMRs. The novel prediction index used in this study is a modified version of the Jaccard index (JI). The traditional JI is a statistical value for comparing the similarity and diversity between two different sample sets. It measures the similarity between two limited sample groups. The value is directly proportional to the similarity between the two groups [6]. Studies have demonstrated that it is more effective to consider the weighted coefficients on similarity analysis for specific problems [7].

To enhance the traditional JI indicator and the evidence of different populations with an identical disease possessing high comorbidity similarities, we proposed a novel weighted JI (WJI) that can effectively reflect the comorbidity distribution of EMRs, instead of using the binary status of comorbidity; further, compared with traditional approaches, we formulated indices to provide accurate prediction results. The details and experimental results are described in the following sections.

*Electronic Medical Records*

EMRs are medical history of patients systematically collected by hospitals and/or insurance institutions; they contain general information on clinical practices, such as medical diagnoses and associated treatments. Wide usage of EMRs can reduce healthcare expenses and medical errors and can improve patients' health [8]. An EMR system has the potential to improve healthcare delivery and to reduce medical costs by enhancing data management capabilities and by mining valuable information from the comprehensive clinical practice database [9]. According to research reports, EMRs play an important role in medical services, such as clinical decision-making support, medical quality monitoring, disease prediction model construction, clinical trial analysis, and treatment personalization.

There are many reports on the different applications of EMRs. Several studies have focused on expanding the clinical contexts of genomic diversities based on EMR-linked genetic data. Denny et al. revealed the associations among rare diseases and genetic effects in relation to prognosis, treatments, drug responses, and comorbidity risk [10]. Based on EMRs, other studies have identified patterns among multiple comorbidities and defined non-random associations between diseases; using different methodologies for feature analysis and similarity detection among these patterns, they found various comorbidity modes [11]. Freund et al. proposed predictive models based on historical insurance claim data to identify and explore patterns of multimorbidity in new customers to detect the high risk of future hospitalizations for constructing intervention management for primary care [12]. Kirchberger et al. identified patterns of comorbidity and multimorbidity by analyzing prevalence figures, logistic regression models, and exploratory factor analysis [13]. The objective of this study was to explore patterns of comorbidity and multimorbidity in the elderly population using different analytical approaches. The report confirmed the existence of certain co-occurring diseases in elderly persons that were not caused by chance.

In some studies, a prediction model featuring patient similarities based on a large amount of medical data was constructed. Different approaches are used for evaluating patient similarities and clustering patient groups, such as analyzing the similarities or distinguishing characteristics among a variety of feature components from patient data. The commonly used Euclidean vector adopts the cosine angle, also known as the cosine similarity measure, between two patients' feature vectors to define the associated patient similarity metric [14]. For example, Tashkandi et al. and Lee et al. evaluated the patient similarities in the ICU dataset (MIMIC-III) using the cosine-similarity-based patient similarity measure (PSM) [15,16].

Taiwan has an internationally well-known National Health Insurance Research Database (NHIRD) [17] that maintains general information of clinical practices in all hospital clinics and integrates the medical records of all Taiwanese citizens. Using this database, researchers can integrate, extract, and convert long-term historical longitudinal medical records from multi-aspect and multi-function databases into a required specification format according to a patient's diagnosis records, medication, and hospitalization information for specific medical applications. In this study, the similarities between the disease records of the control and experimental groups were calculated to identify those with high-risk factors through EMRs. In the following sections, we describe the construction of the prediction model using the proposed WJI and verify the effectiveness of the novel proposed indicator.

## 2. Materials and Methods

### 2.1. Source of Materials

The data used in this study were anonymous medical data authorized by the NHIRD (Taiwan), consisting of one million insured people collected between 1996 and 2013 (IRB: 104-5430B). The disease classification code followed the international disease classification standard ICD-9-CM, which contains 17 chapters and 2 supplementary categories. The 17 major chapters can be further labeled and classified into 143 mid-level classification and 999 individual disease classes. The information used in this study was mainly based on an analysis of the disease codes at the individual- and the mid-levels. According to the definition of the disease group, the individual-level classification was represented by a three-digit code for a single disease, and each mid-level classification represented a disease group. Hence, 999 disease categories were defined based on individual-level classification; these individual disease categories were grouped into 143 disease groups, which were defined as mid-level classification groups.

### 2.2. Medical Histories and Feature Extraction Analysis

Firstly, positive and negative data groups were defined. The positive and negative data groups were composed of subjects with and without ALS, respectively. The positive subjects were obtained directly from the one million NHIRD medical database, and the relatively larger number of negative subjects were retrieved from the same database based on matching gender and age attributes according to the positive subjects.

The experimental data obtained from the NHIRD database underwent data cleaning and data integration to yield two complete sets of medical diagnosis records for both groups. Furthermore, the subjects for the different disease codes for each group were counted. Statistical analysis was performed on the different disease codes to select associated disease comorbidities for both the experimental and control groups for constructing the prediction systems.

Figure 1a,b are simple schematic diagrams of the disease set corresponding to the experimental and control groups. These two diagrams were designed for illustration; they do not represent actual medical data contents. The data represent the comprehensive disease records of the positive subjects (diagnosed with the ALS disease) within two years before diagnosis. The total number of subjects suffering from a specific disease in the experimental or control group was noted, and the comorbidity codes were preliminarily screened by evaluating a minimum support threshold setting. A minimum support threshold value

represents the minimum number of subjects or the minimum percentage of total subjects within the experimental group. In other words, comorbidities which were present in less than a certain percentage of subjects within the experimental group were discarded from the feature set in this study.

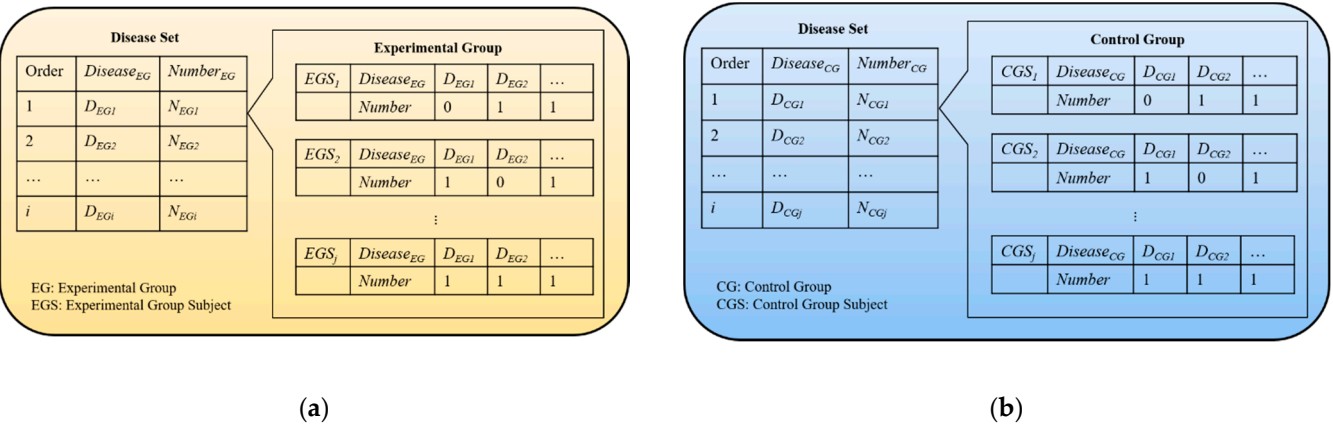

**(a)**                                                                 **(b)**

**Figure 1.** Illustration of the disease set from (**a**) the experimental group and (**b**) the control group: on the right part of the two tables are disease records of each subject, and each disease is displayed within the defined interval binary representation of 0 and 1. The left side is the disease set of the experimental/control group and the total number of subjects suffering various diseases. $Disease_{EG}$ and $Disease_{CG}$ refer to disease classification codes in the experimental and control groups, respectively. $Number_{EG}$ and $Number_{CG}$ refer to the total number of subjects in the experimental and control groups suffering from a certain disease. $D_{EGi}$ and $D_{CGi}$ refer to the $i^{th}$ disease classification code in the experimental and control groups, respectively. $N_{EGi}$ and $N_{CGi}$ refer to the total number of subjects in the experimental/control group suffering from the $i^{th}$ disease under a classification code. $EGS_j$ and $CGS_j$ refer to the $j^{th}$ subject in the experimental and control groups, respectively.

Figure 2 is a schematic diagram of the comorbidity feature set. For the disease code sets from both the experimental and control groups, the same lead-time interval and an identical minimum support threshold setting were applied to define the associated disease records for comorbidity analysis. To identify highly associated comorbidities with ALS, the classification codes of all the comorbidities were further analyzed by evaluating the corresponding odds ratios (ORs) for ALS. In other words, the OR of each comorbidity was calculated to evaluate the chances of a comorbidity associated with ALS. A set of comorbidity codes with ORs > 2 was constructed and defined as an associated comorbidity feature set for ALS. Since the associated comorbidity set of ALS was retrieved from the disease set of the original experimental group, after the OR analysis, the selected comorbidities must be a subset from the disease set of the experimental group. Therefore, the number of subjects for the filtered associated diseases in the associated comorbidity feature set should be the same as that of the subjects within the original experimental group. This process may remove background noise (unassociated comorbidities) from the constructed prediction model. For example, the disease code sets of both the experimental and control groups may simultaneously contain a seasonal cold; hence, a small OR could be obtained and no specific correlation was detected between seasonal cold and ALS. Thus, following the statistical significance analysis of all comorbidities, the system defined a comorbidity feature set with strong association with ALS, which can explain the strong relationship between the comorbidity codes and ALS.

| Order | $Disease_{FS}$ | $Number_{EG}$ |
|-------|----------------|----------------|
| 1 | $D_{FS1}$ | $N_{EG1}$ |
| 2 | $D_{FS2}$ | $N_{EG2}$ |
| … | … | … |
| $i$ | $D_{FSi}$ | $N_{EGi}$ |

EG: Experimental Group
FS: Comorbid Disease Feature Set

**Figure 2.** Illustration of the comorbidity feature set: $Disease_{FS}$ refers to comorbidity classification codes with odds ratios > 2. $Number_{EG}$ refers to the total number of subjects in the experimental group suffering from a specific disease. The number of subjects suffering from a specific disease in the comorbidity feature set is the same as the number of subjects in the disease set in the experimental group ($Number_{EG}$).

### 2.3. Proposed Patient Prediction Module

The proposed diagnostic model is mainly based on the similarity measurement of comorbidity patterns between testing subjects and known ALS patients. From the experimental and control data groups, we initially calculated the odds ratios for each comorbidity (within a two-year interval) for measuring the association between comorbidities and ALS disease. Based on odds ratio analysis, comorbidities with strong associations were selected and constructed as an important comorbidity feature set, and the proposed WJI similarity measurements were further calculated for the experimental group vs. the comorbidity feature set and for the control group vs. the comorbidity feature set. A binary outcome prediction model for supporting clinical decision making was trained on the calculated WJI indicators of the experimental (ALS) and control (non-ALS) groups versus the ALS-associated comorbidity feature set. Once the WJI indicators of the experimental and control data groups were trained and constructed as a prediction model, only the selected associated comorbidities of testing subjects were considered and evaluated according to the WJI similarity measurement and compared to the previously trained thresholding value. The ALS prediction module was constructed and described as the following. Obtaining the comorbidities for each subject in the experimental or control groups, and its corresponding WJI similarity was calculated with respect to the defined associated comorbidity feature set. Subsequently, a Z-score standardization procedure was performed for similarity measurement and for training and construction of the probability prediction model. Excluding the subjects in the experimental and control data groups, for which the data were utilized as the training dataset (80% of samples), the data of the remaining subjects were applied as the testing data (20% of samples).

In this study, four different machine learning models were utilized for data training: Logistic Regression (LR) [18], Support Vector Classifier (SVC) [19,20], random forest [21], and eXtreme Gradient Boosting (XGBoost) [22]; five-fold cross validation was adopted to verify the accuracy and prediction stability of the trained models. All four machine learning techniques are discrimination supervised learning models that learn from the training dataset and build a model to make predictions for unseen data in different classification applications. Both the LR and SVC are prediction models commonly used in traditional medical applications. The random forest and XGBoost, which are the most popular machine learning models in recent years, are both ensemble learning approaches designed by integrating a variety of learning algorithms to achieve better prediction capabilities.

### 2.4. Jaccard Index

The traditional Jaccard index (JI) defines the similarity between two different sample sets, *A* and *B*. The index is shown in Equation (1):

$$\text{Jaccard}(A,\ B) = \frac{|A \cap B|}{|A \cup B|} = \frac{|A \cap B|}{|A| + |B| - |A \cap B|} \tag{1}$$

where $|A \cap B|$ represents the number of overlapped items and $|A \cup B|$ represents the number of union items. Regardless of the occurrence frequency of each item (such as disease classification code), "1" and "0" denote existing and nonexisting conditions, respectively. This similarity indicator confers equal importance of co-occurring items.

However, to predict a target disease through the comorbidity analysis, the co-occurrence frequencies of different comorbidities should possess different weights for disease evaluation. For example, pregnant women with high blood pressure-related diseases would have a relatively higher probability of premature delivery than those who suffer from skin diseases. However, the traditional JI can only mark different comorbidity codes as "1" or "0," ignoring the incidence of certain important comorbidities.

To strengthen the occurrence frequency of various comorbidities from the EMRs of known patients, we proposed an improved WJI to replace the traditional JI for better evaluation of the similarities between two comorbidity sets. This novel index utilizes the proportional incident rate for corresponding weight calculation and enhances the accuracy of the similarity measurement between positive and negative training datasets. The WJI-related terminologies are defined in Table 1; several examples are illustrated in the supplementary document I.

**Table 1.** Number of patients and corresponding weights for $AG*$ and $BG*$.

| | | | | | | |
|---|---|---|---|---|---|---|
| | $Disease_A$ | $d_{A1}$ | $d_{A2}$ | ... | $d_{Ai-1}$ | $d_{Ai}$ |
| $AG*$ | $Number_A$ | $N_{A1}$ | $N_{A2}$ | ... | $N_{Ai-1}$ | $N_{Ai}$ |
| | $Weight_A$ | $\frac{N_{A1}}{\sum N_{Ai}}$ | $\frac{N_{A2}}{\sum N_{Ai}}$ | ... | $\frac{N_{Ai-1}}{\sum N_{Ai-1}}$ | $\frac{N_{Ai}}{\sum N_{Ai}}$ |
| | $Disease_B$ | $d_{B1}$ | $d_{B2}$ | ... | $d_{Bj-1}$ | $d_{Bj}$ |
| $BG*$ | $Number_B$ | $N_{B1}$ | $N_{B2}$ | ... | $N_{Bj-1}$ | $N_{Bj}$ |
| | $Weight_B$ | $\frac{N_{B1}}{\sum N_{Bj}}$ | $\frac{N_{B2}}{\sum N_{Bj}}$ | ... | $\frac{N_{Bj-1}}{\sum N_{Bj-1}}$ | $\frac{N_{Bj}}{\sum N_{Bj}}$ |

*AG*: set of comorbidities for patient group A within the defined interval before the patient is diagnosed with specific target disease. *AG*∗: top *i* frequently co-occurring diseases content set of *AG*. *Disease_A*: comorbidities in patient group A. $d_{Ai}$: $i^{th}$ comorbidity in group A. *Number_A*: number of patients with a specific comorbidity in group A. $N_{Ai}$: number of patients with the $i^{th}$ comorbidity in group A. *Weight_A*: corresponding weights of specific comorbidities in group A. $\frac{N_{Ai}}{\sum NA}$: the corresponding weight of the $i^{th}$ specific comorbidity in group A. *BG*: the comorbidity records for group B within the same interval as *AG*. *BG*∗: top *j* frequently co-occurring diseases content set of *BG*. *Disease_B*: comorbidities in patient group B. $d_{Bj}$: $j^{th}$ comorbidity in group B. *Number_B*: number of patients with a specific comorbidity in group B. $N_{Bj}$: number of patients with the $j^{th}$ comorbidity in group B. *Weight_B*: corresponding weights of specific comorbidities in group B. $\frac{N_{Bj}}{\sum NB}$: corresponding weight of the $j^{th}$ specific comorbidity in group B.

There are two different types of comorbidities, and different methods are used to calculate the corresponding WJIs. For the first type, the comorbidity classification codes appeared exclusively in either set $AG*$ or $BG*$. When a comorbidity occurred only in $AG*$, the corresponding weighting coefficient for the specific disease code was obtained by dividing the number of patients with such a comorbidity within $AG*$ by the total number of patients with each type of comorbidity in $AG*$. Similarly, the weighting coefficient algorithm can be applied to any single comorbidity occurring in $BG*$ exclusively.

In contrast, the second type of comorbidity code occurred within $AG*$ and $BG*$ simultaneously. In this case, the corresponding weights of the co-occurring comorbidity codes were calculated by taking proportional measurements in each patient group as the

first type and then by taking an average from both $AG*$ or $BG*$ as the final corresponding weights for the common comorbidities. The formula is denoted as Equation (2):

$$Weight = \begin{cases} \frac{N_{Ai}}{\sum N_{Ai}} \text{ or } \frac{N_{Bi}}{\sum N_{Bi}}, \text{ if } d_{Ai} \neq d_{Bj} \\ \frac{\frac{N_{Ai}}{\sum N_{Ai}} + \frac{N_{Bj}}{\sum N_{Bj}}}{2}, \text{ if } d_{Ai} = d_{Bj} \end{cases} \quad (2)$$

The WJI was then obtained by summing up all the weighting coefficients corresponding to $AG*$ and $BG*$ of the commonly occurring comorbidity codes and then by dividing by the sum of all the weights of the disease classification codes within $AG*$ and $BG*$. A few examples are given in the Supplementary Materials.

### 2.5. Scoring Methods

To evaluate the proposed WJI similarity-based ALS prediction system, two different approaches including the traditional JI similarity (without weighting concern) and the scoring mechanism (score) were constructed for comparison. The scoring method is similar to the symptom indexing chart used in the past for the initial diagnosis of a specific disease. Different approaches were used for estimating the risk of suffering a target disease. A common approach was to design a score sheet for evaluating disease symptoms. Another common approach was to administer questionnaires. For example, the International Prostate Symptom Score, the Edinburgh Postnatal Depression Scale, and kidney disease include scores sheet to evaluate associated symptoms [23–25]

In the following experimental analysis, the scoring index was defined by taking the comorbidity classification codes as the symptom feature set; subsequently, a related weighted proportional score was given according to the historical number of patients. Finally, the weighted scores of the comorbidity records of the test subject within a specific duration were summed up, and the score for the test subject was obtained. The score acts as a preliminary prediction of the possibility of the subject suffering the disease in the near future. The simple illustration in Table 2 shows the conception of the intuitive scoring mechanism defined in this study.

**Table 2.** Simple illustration of scoring mechanism.

| $Disease_{FS}$ | $Number_{FS}$ | Weight | Patient Score |
|---|---|---|---|
| $d_{A1}$ | 4 | $\frac{4}{20} = 0.20$ | V |
| $d_{A2}$ | 6 | $\frac{6}{20} = 0.30$ | |
| $d_{A3}$ | 10 | $\frac{10}{20} = 0.50$ | V |
| Total | 20 | 1 | $0.20 + 0.50 = 0.70$ |

The left side is the disease classification codes and the total number of subjects in the comorbidity feature set; the related weighted proportional score was given according to the historical number of patients. The right side is the final score of the test subject, which is the sum of the weighted scores of comorbidity records of a subject within a specific period. $Disease_{FS}$ is the disease classification codes in the comorbidity feature set. $Number_{FS}$ is the total number of subjects in the comorbidity feature set suffering from a certain disease. $Weight$ is the corresponding weights of specific comorbidities. $Patient Score$ is the final score of a test subject.

### 2.6. Amyotrophic Lateral Sclerosis (ALS)

In this study, a total of 162 ALS patients were identified, and their corresponding EMRs were retrieved from the selected governmental medical database composed of one million data (IRB: 104-5430B). To confirm that the initial subjects were indeed ALS patients (excluding suspected cases), we only considered patients who were hospitalized due to the disease. Hence, only 71 subjects were selected for this study (male–female ratio = 45:26).

Among them, 41 were hospitalized directly upon diagnosis; the remaining 30 subjects were hospitalized after some time following the diagnosis. The average period of hospitalization for ALS following the first diagnosis was 2 years.

The 71 confirmed subjects were considered the experimental group, and their historical disease records within 2 years of diagnosis were defined as *ALS_EG*. The total number of subjects suffering from each disease was recorded as the disease set of the experimental group, and the comorbidity classification codes were preliminarily screened and defined as *ALS_EG_FEA*. To further analyze and compare the different parameter settings, the comorbidity classification code of *ALS_EG_FEA* that was found among more than 10% of the total number of patients (71) was extracted and defined as *ALS_EG_FEA$_{0.1}$*.

The subjects of the control group were retrieved from the database based on the age and ratio of each gender. A total number of 399 subjects without the ALS disease were selected as the control group (male–female ratio = 249:150). According to the governmental database in Taiwan, the average onset age of the retrieved ALS subjects is 51 years old. Therefore, the historical medical records between 49 and 51 years old were retrieved for each control group subject and defined as *ALS_CG*. All the various disease records for the control group were enumerated and counted, and the corresponding comorbidity codes were preliminarily screened and defined as *ALS_CG_FEA*. To analyze and compare the different parameter settings, a comorbidity code occurring among more than 10% of the 399 subjects was additionally extracted and defined as *ALS_CG_FEA$_{0.1}$*.

Finally, we calculated the odds for each comorbidity code within both *ALS_EG_FEA* and *ALS_CG_FEA* and divided them to obtain an OR for explaining the strength of specific comorbidities associated with ALS. In other words, the OR identifying a specific comorbidity was calculated. When OR > 2, the associated disease code was selected and defined as an important feature for subsequent identification; the comorbidity set collected was defined as *ALS_SELECT_FEA*. These associated comorbidities were defined as the feature set of the ALS comorbidities that were used to explain the important correlation between certain comorbidities and ALS diseases. Further, to select a feature disease set with stronger representation, a screening condition based on a proportion greater than 10% of the total subjects was established and defined as *ALS_EG_FEA$_{0.1}$* and *ALS_CG_FEA$_{0.1}$*. Similar procedures for calculating the ORs for each disease code from *ALS_EG_FEA$_{0.1}$* and for selecting comorbidities with ORs > 2 were performed for subsequent identification, and the comorbidity set was defined as *ALS_SELECT_FEA$_{0.1}$*.

Because there were two different comorbidity feature sets, it was necessary to construct two different ALS prediction models for at-risk patient recognition. The first prediction system focused on each subject within *ALS_EG* and *ALS_CG* and adopted the WJI similarity analysis, with respect to *ALS_SELECT_FEA*. The similarity between each subject within the comorbidity feature set was calculated and normalized, following which a probability prediction model was constructed. Subsequently, the model was used to predict the probability of a subject suffering from ALS. Finally, five-fold cross validation was adopted to verify the accuracy and prediction stability of the trained model. In contrast, the second prediction system used a disease occurrence rate >10%, that is, *ALS_SELECT_FEA$_{0.1}$*, which was obtained using *ALS_EG_FEA$_{0.1}$* and *ALS_CG_FEA$_{0.1}$* for significance analysis. The final prediction results were obtained by analyzing each subject of *ALS_EG* and *ALS_CG* using *ALS_SELECT_FEA$_{0.1}$*.

*2.7. Model Evaluation Index*

The model score, precision, sensitivity, specificity, accuracy, F1 score, and area under the receiver operating characteristic curve (AUC) were utilized as the evaluation indicators for comparing different predictive models [26]. A true positive (TP) is the number of subjects in the positive class correctly predicted as belonging to the positive class; a true negative (TN) is the number of subjects in the negative class correctly predicted as belonging to the negative class; a false positive (FP) is the number of subjects in the negative class incorrectly predicted as belonging to the positive class; and a false negative (FN) is the

number of subjects in the positive class incorrectly predicted as belonging to the negative class.

Precision is defined as TP/(TP+FP).

Sensitivity, also referred to as the recall rate, is defined as TP/(TP+FN).

Specificity is defined as TN/(FP+TN);

Accuracy was the accuracy rate of prediction based on the testing data; it is defined as (TP+TN)/(TP+FP+TN+FN); the model score is the accuracy rate of prediction obtained from the training data itself.

F1 score refers to the harmonic average of the precision and the recall rate.

AUC refers to the area under the receiver operating characteristic curve, which represents a statistical value of the predictive ability of the classifier; the larger its value, the better the performance of the prediction model.

## 3. Results and Discussion

We analyzed comorbidities of ALS disease with a strong association through odds ratio analysis. The selected comorbidities were categorized into the individual level and mid-level. Both selected comorbidities with strong associations in different levels are shown in the Supplementary Materials. We also compared pair similarities among the disease set of the experimental group vs. control group and thee comorbidity feature set vs. control group. The value of JI similarity between the disease set of the experimental group and control group is always larger than the JI similarity between the comorbidity feature set and control group. It is due to the comorbidity feature set removing a lot of unassociated diseases by statistical analytics. Hence, several common diseases such as the seasonal flu and common cold were removed and the similarity indicator decreased. Similarly, the similarity measurement for the WJI indicator possesses the same decreasing trend. From Table 3, it can be observed that the differentiation value of JI indicator (0.22) is less than the WJI indicator (0.43). It is mainly due to each disease code in the feature set weighted in the WJI indicator according to the disease population, and the difference was enlarged due to unevenly distributed subjects. Therefore, the WJI indicator provided a better differentiating factor than the JI indicator in general. It is an important property to provide a better prediction performance for discriminating ALS patients and health subjects. The similarities in the individual disease and mid-level clustered disease code levels among *ALS_EG_FEA*, *ALS_CG_FEA*, and *ALS_SELECT_FEA* are shown in Tables 3 and 4. For different disease code levels (either individual or mid-level codes), the differentiation values also preserved an identical trend between the WJI and JI indicators.

**Table 3.** Jaccard index and weighted Jaccard index similarities pairwise *ALS_EG_FEA*, *ALS_CG_FEA*, and *ALS_SELECT_FEA* (individual-level classification).

| Individual-Level Classification | Jaccard Index | Weighted Jaccard Index |
|---|---|---|
| | Disease Set from Control Group [2] | Disease Set from Control Group [2] |
| Disease Set from Experimental Group [1] | 0.55 | 0.87 |
| Comorbidity Feature Set [3] | 0.33 | 0.44 |
| Differentiation | 0.22 | 0.43 |

[1] Disease set from the experimental group: *ALS_EG_FEA*; [2] Disease set from the control group: *ALS_CG_FEA*; [3] Comorbidity feature set: *ALS_SELECT_FEA*.

**Table 4.** Jaccard index and weighted Jaccard index similarities pairwise *ALS_EG_FEA*, *ALS_CG_FEA*, and *ALS_SELECT_FEA* (mid-level classification).

| Mid-Level Classification | Jaccard Index | Weighted Jaccard Index |
|---|---|---|
| | Disease Set from Control Group [2] | Disease Set from Control Group [2] |
| Disease Set from Experimental Group [1] | 0.75 | 0.98 |
| Comorbidity Feature Set [3] | 0.43 | 0.53 |
| Differentiation | 0.32 | 0.45 |

[1] Disease set from experimental group: *ALS_EG_FEA*; [2] Disease set from control group: *ALS_CG_FEA*; [3] Comorbidity feature set: *ALS_SELECT_FEA*.

Tables 5 and 6 show the pair similarities of the individual disease code level and middle-clustered disease code level among constrained disease feature groups ($ALS\_EG\_FEA_{0.1}$ vs. $ALS\_CG\_FEA_{0.1}$, and $ALS\_SELECT\_FEA_{0.1}$ vs. $ALS\_CG\_FEA_{0.1}$). From Tables 3–6, the differences in the WJI between the disease set of the experimental and control groups are higher than those of the JI. Again, the main reason for this is that the WJI involves weight coefficient calculations and possesses more detailed associations than the JI. When the number of subjects suffering from an identical disease between the two sets was large, the relative weight coefficient for the specific comorbidity and the degree of influence became associated. However, when the number of subjects suffering from diseases under various classification codes was large but the number of subjects was small, the relative weight coefficient for the specific comorbidity was reduced and the corresponding impact decreased. The traditional JI similarity is based solely on the presence or absence of a disease classification code; it does not consider the difference in the prevalence of the associated comorbidity codes. The binary counting of 0 and 1 was used to analyze the comorbidity relationship. Therefore, the value of the WJI provides better distinguishability than that using JI.

**Table 5.** Jaccard index and weighted Jaccard index similarities pairwise $ALS\_EG\_FEA_{0.1}$, $ALS\_CG\_FEA_{0.1}$, and $ALS\_SELECT\_FEA_{0.1}$ (individual-level classification).

| Individual-Level Classification | Jaccard Index | Weighted Jaccard Index |
|---|---|---|
| | Disease Set from Control Group [2] | Disease Set from Control Group [2] |
| Disease Set from Experimental Group [1] | 0.48 | 0.65 |
| Comorbidity Feature Set [3] | 0.19 | 0.23 |
| Differentiation | 0.29 | 0.42 |

[1] Disease set from the experimental group: $ALS\_EG\_FEA_{0.1}$; [2] Disease set from control group: $ALS\_CG\_FEA_{0.1}$; [3] Comorbidity feature set: $ALS\_SELECT\_FEA_{0.1}$.

A high similarity between the comorbidity set of the experimental and control groups implies a small difference between the two sets. This might be due to too many common comorbidity codes within the similarity analysis. For example, seasonal flu is a common disease in both groups; therefore, this disease code has no association in distinguishing between both two groups. Hence, we used statistical verification to eliminate disease codes that failed to satisfy the significance analysis. A new feature set with the associated disease codes was constructed for comparison, which showed a larger difference between the associated feature set and the original disease code set of the control group. The tables show that the difference in the similarities according to the WJI was larger than that using the JI. Therefore, through our proposed method, two original feature sets with

no discriminative attributes could be improved by constructing the associated feature set to realize a better prediction model. The similarity measurement according to the middle-clustered classification codes was generally higher than that of the individual-level classification; this is mainly because the individual-level classification representing each disease code was a specific disease, and the mid-level clustered classification implied that each code represents a group of similar diseases. A disease group contains many individual disease codes, which increase the similarity measurement in relation to the middle-clustered classification models.

**Table 6.** Jaccard index and weighted Jaccard index similarities pairwise $ALS\_EG\_FEA_{0.1}$, $ALS\_CG\_FEA_{0.1}$, and $ALS\_SELECT\_FEA_{0.1}$ (mid-level classification).

| Mid-Level Classification | Jaccard Index | Weighted Jaccard Index |
|---|---|---|
| | Disease Set from Control Group [2] | Disease Set from Control Group [2] |
| Disease Set from Experimental Group [1] | 0.61 | 0.73 |
| Comorbidity Feature Set [3] | 0.27 | 0.34 |
| Differentiation | 0.34 | 0.39 |

[1] Disease set from the experimental group: $ALS\_EG\_FEA_{0.1}$; [2] Disease set from the control group: $ALS\_CG\_FEA_{0.1}$; [3] Comorbidity feature set: $ALS\_SELECT\_FEA_{0.1}$.

According to the individual-level and middle-clustered classification of disease codes and different parameter settings, to construct the training data and prediction system, each subject of *ALS_EG* and *ALS_CG* was analyzed based on the similarities between two different comorbidities. The verification of the prediction results is shown in Tables 7–10.

Tables 7 and 8 show the prediction results obtained by training *ALS_EG*, *ALS_CG*, and *ALS_SELECT_FEA*. In addition to the XGBoost, compared with the JI and traditional scoring analysis, the WJI improved the performance of the training models and prediction results. Table 7 shows the prediction results obtained using individual disease code-level classification for model training. Excluding the XGBoost, the prediction results of the other three machine learning models based on the WJI was higher than that of the JI and traditional scoring analysis in terms of sensitivity, accuracy, F1 score, and AUC value. The results obtained by applying the middle-clustered classification for model training are shown in Table 8. In addition to the XGBoost, all the training and prediction results of the other three machine learning models based on the WJI outperformed the JI and traditional scoring in terms of model score, precision, sensitivity, specificity, accuracy, F1 score, and AUC value.

According to the WJI training verification results shown in Tables 7 and 8, the performance of the XGBoost based on the individual disease code-level classification was generally worse than that of the mid-level clustered classification, unlike the other three machine learning models. The comparisons between the individual- and mid-level clustered disease codes for the three machine learning models were analyzed as follows. In the mid-level clustered classification, the models trained using the SVC and random forest approach could achieve a high accuracy of 0.808, sensitivity of 0.729, specificity of 0.886, and AUC value of 0.844. However, the individual-level classification outperformed the mid-level clustered classification. The SVC and random forest models achieved an accuracy of 0.837, sensitivity of 0.788, specificity of 0.857, and AUC values of 0.907.

**Table 7.** Training and verification results of *ALS_EG*, *ALS_CG* and *ALS_SELECT_FEA* (individual level classification).

| Individual Level Classification (Mean of 5-fold Values) | Logistic Regression | | | Support Vector Classification | | | Random Forest | | | XGBoost | | |
|---|---|---|---|---|---|---|---|---|---|---|---|---|
| | JI [1] | WJI [2] | Score | JI | WJI | Score | JI | WJI | Score | JI | WJI | Score |
| Model Score | 0.785 | 0.840 | 0.753 | 0.662 | 0.852 | 0.715 | 0.805 | 0.866 | 0.776 | 0.949 | 0.977 | 0.933 |
| Precision | 0.883 | 0.854 | 0.824 | 1.0 | 0.87 | 0.823 | 0.84 | 0.883 | 0.784 | 0.835 | 0.79 | 0.643 |
| Sensitivity | 0.659 | 0.83 | 0.659 | 0.322 | 0.816 | 0.491 | 0.688 | 0.788 | 0.688 | 0.816 | 0.745 | 0.562 |
| Specificity | 0.915 | 0.843 | 0.859 | 1.0 | 0.857 | 0.9 | 0.872 | 0.886 | 0.816 | 0.814 | 0.787 | 0.704 |
| Accuracy | 0.787 | **0.836** | 0.759 | 0.661 | **0.837** | 0.695 | 0.78 | **0.837** | 0.752 | **0.815** | 0.766 | 0.633 |
| F1 Score | 0.751 | 0.836 | 0.725 | 0.481 | 0.835 | 0.613 | 0.752 | 0.829 | 0.729 | 0.817 | 0.755 | 0.596 |
| AUC | 0.872 | 0.907 | 0.816 | 0.411 | 0.907 | 0.816 | 0.87 | 0.915 | 0.807 | 0.815 | 0.766 | 0.633 |

[1] JI represents Jaccard index; [2] WJI represents weighted Jaccard index.

**Table 8.** Training and verification results of *ALS_EG*, *ALS_CG* and *ALS_SELECT_FEA* (mid-level classification).

| Mid-Level Classification (Mean of 5-fold Values) | Logistic Regression | | | Support Vector Classification | | | Random Forest | | | XGBoost | | |
|---|---|---|---|---|---|---|---|---|---|---|---|---|
| | JI | WJI | Score | JI | WJI | Score | JI | WJI | Score | JI | WJI | Score |
| Model Score | 0.739 | 0.792 | 0.743 | 0.755 | 0.812 | 0.743 | 0.774 | 0.828 | 0.751 | 0.924 | 0.977 | 0.921 |
| Precision | 0.762 | 0.787 | 0.733 | 0.77 | 0.866 | 0.699 | 0.794 | 0.865 | 0.694 | 0.827 | 0.784 | 0.65 |
| Sensitivity | 0.716 | 0.801 | 0.728 | 0.659 | 0.731 | 0.728 | 0.659 | 0.729 | 0.703 | 0.772 | 0.788 | 0.659 |
| Specificity | 0.761 | 0.774 | 0.733 | 0.789 | 0.886 | 0.689 | 0.801 | 0.887 | 0.689 | 0.818 | 0.774 | 0.635 |
| Accuracy | 0.739 | **0.788** | 0.731 | 0.724 | **0.808** | 0.709 | 0.73 | **0.808** | 0.696 | **0.795** | 0.781 | 0.647 |
| F1 Score | 0.731 | 0.788 | 0.723 | 0.703 | 0.791 | 0.707 | 0.71 | 0.787 | 0.695 | 0.789 | 0.784 | 0.649 |
| AUC | 0.837 | 0.876 | 0.812 | 0.816 | 0.844 | 0.794 | 0.833 | 0.866 | 0.788 | 0.795 | 0.781 | 0.647 |

**Table 9.** Training and verification results of *ALS_EG*, *ALS_CG* and *ALS_SELECT_FEA$_{0.1}$* (individual level classification).

| Individual Level Classification (Mean of 5-fold Values) | Logistic Regression | | | Support Vector Classification | | | Random Forest | | | XGBoost | | |
|---|---|---|---|---|---|---|---|---|---|---|---|---|
| | JI | WJI | Score | JI | WJI | Score | JI | WJI | Score | JI | WJI | Score |
| Model Score | 0.753 | 0.785 | 0.746 | 0.75 | 0.776 | 0.736 | 0.759 | 0.794 | 0.759 | 0.912 | 0.963 | 0.919 |
| Precision | 0.76 | 0.789 | 0.754 | 0.76 | 0.78 | 0.739 | 0.752 | 0.795 | 0.794 | 0.698 | 0.765 | 0.733 |
| Sensitivity | 0.744 | 0.773 | 0.715 | 0.744 | 0.719 | 0.715 | 0.702 | 0.759 | 0.688 | 0.673 | 0.773 | 0.676 |
| Specificity | 0.761 | 0.788 | 0.747 | 0.761 | 0.787 | 0.732 | 0.76 | 0.802 | 0.817 | 0.718 | 0.761 | 0.731 |
| Accuracy | 0.752 | **0.781** | 0.731 | 0.752 | **0.753** | 0.724 | 0.731 | **0.781** | 0.752 | 0.696 | **0.767** | 0.704 |
| F1 Score | 0.747 | 0.778 | 0.724 | 0.747 | 0.742 | 0.718 | 0.721 | 0.775 | 0.731 | 0.679 | 0.768 | 0.698 |
| AUC | 0.821 | 0.832 | 0.8 | 0.796 | 0.809 | 0.773 | 0.811 | 0.841 | 0.797 | 0.696 | 0.767 | 0.704 |

**Table 10.** Training and verification results of *ALS_EG*, *ALS_CG* and *ALS_SELECT_FEA$_{0.1}$* (mid-level classification).

| Mid-Level Classification (Mean of 5-fold Values) | Logistic Regression | | | Support Vector Classification | | | Random Forest | | | XGBoost | | |
|---|---|---|---|---|---|---|---|---|---|---|---|---|
| | JI | WJI | Score | JI | WJI | Score | JI | WJI | Score | JI | WJI | Score |
| Model Score | 0.738 | 0.75 | 0.734 | 0.746 | 0.767 | 0.743 | 0.767 | 0.781 | 0.746 | 0.901 | 0.977 | 0.896 |
| Precision | 0.744 | 0.735 | 0.733 | 0.762 | 0.812 | 0.684 | 0.769 | 0.786 | 0.683 | 0.769 | 0.713 | 0.55 |
| Sensitivity | 0.728 | 0.729 | 0.728 | 0.687 | 0.702 | 0.714 | 0.617 | 0.689 | 0.674 | 0.733 | 0.731 | 0.503 |
| Specificity | 0.747 | 0.746 | 0.733 | 0.788 | 0.829 | 0.676 | 0.815 | 0.8 | 0.689 | 0.774 | 0.704 | 0.605 |
| Accuracy | **0.738** | 0.737 | 0.731 | 0.737 | **0.765** | 0.695 | 0.716 | **0.744** | 0.682 | **0.754** | 0.718 | 0.554 |
| F1 Score | 0.729 | 0.724 | 0.723 | 0.716 | 0.746 | 0.69 | 0.68 | 0.726 | 0.675 | 0.748 | 0.72 | 0.523 |
| AUC | 0.83 | 0.846 | 0.806 | 0.799 | 0.807 | 0.79 | 0.831 | 0.843 | 0.783 | 0.754 | 0.718 | 0.554 |

Tables 9 and 10 show the various verification results for *ALS_EG*, *ALS_CG*, and *ALS_SELECT_FEA$_{0.1}$*. It can be observed that the four machine learning models based on the WJI generally outperformed the JI and traditional scoring analysis. Table 9 is the prediction result obtained from using the individual disease codes for model training. For model training, the four different machine learning models incorporating the WJI yielded scores better in terms of precision, accuracy, and AUC values compared with the JI and traditional scoring analysis. Table 10 shows the results obtained using the middle-clustered-level classification for model training. The SVC and random forest learning models based on the WJI, rather than the JI and traditional scoring indicator, achieved better performance.

The comparison of the performance of the WJI-based training models for the individual disease code and the middle clustered-level classification for the four different machine learning models are shown in Tables 9 and 10. For the mid-level clustered classification, the SVC learning model achieved a high accuracy of 0.765, sensitivity of 0.702, specificity of 0.829, and AUC value of 0.807. However, the individual-level classification was more effective than the middle-clustered-level classification. Among the four different learning models, the LR and random forest achieved an accuracy of 0.781, sensitivity of 0.759, specificity of 0.788, and AUC value of 0.832.

Tables 7–10 show that the learning models based on the WJI generally outperformed the ones based on the JI and traditional scoring analysis. Specifically, Tables 7 and 8 show that the prediction results for each subject in *ALS_EG* and *ALS_CG* against *ALS_SELECT_FEA* were better than that in *ALS_SELECT_FEA$_{0.1}$*. In addition, the results obtained at the individual disease code level (shown in Table 7) were better than those of the mid-level clustered classification (Table 8). Because each mid-level clustered code represented a group of similar diseases, the corresponding weighted coefficients might be diluted during feature training, resulting in reduced effectiveness in distinguishing the various comorbidity features. This leads to poor performance during model training and cross-verification prediction.

In this study, ALS was applied as the target disease for validating the proposed method. As a matter of fact, different multivariate models were considered initially, but the performance was not good and not reported in this study. It is believed that the many comorbidities and few ALS subjects were the main reasons causing unsatisfactory results. Therefore, we proposed the novel similarity-based approach instead of the statistical modeling approach. To validate the proposed efficient and effective prediction systems, four different machine learning models were adopted. To further analyze the prediction models with different parameter settings, two different levels of comorbidity, namely individual disease codes and mid-level clustered disease codes, were applied for constructing the prediction model and for comparing the system performance. To enhance the effectiveness of the proposed model, in addition to the proposed WJI similarity, the traditional JI similarity and a scoring mechanism were applied.

According to our survey results, our proposed machine learning method using WJI indicators is the first approach based on Electronic Medical Records for ALS prediction. Previous ALS diagnosis mainly focused on clinical biomarkers, biological biomarkers, genetic/proteomics biomarkers, and neuroimaging indicators [27]. Several machine learning techniques have been extensively applied to assist ALS diagnosis using the mentioned biomarkers. Among all different biomarker usages, using a neuroimaging approach achieved the best performance of higher than 80% sensitivity in general [28,29]. However, these diagnostic approaches often occurred after serious symptom onset and led to diagnostic delay. How to apply noninvasive approaches with outstanding sensitivities and specificities at an early stage has become an important issue for ALS patients, and such a diagnostic model could help at-risk ALS patients to be recruited into earlier clinical trials. One advantage of our proposed method is that using individual EMRs as features to compare with comorbidity patterns of ALS patients is a noninvasive, cost-free, and efficient approach for early detection. However, no single test could provide a definitive diagnosis of ALS so far. Several diseases such as multiple sclerosis and Parkinson's disease hold simi-

lar symptoms to ALS, and these symptoms may be initially neglected by non-ALS-trained physicians. Hence, the proposed approach only provides an early alert for physicians. Complete neurologic examinations for muscle weakness, spasticity, and atrophy are still required for precise diagnosis of ALS.

Through historical disease records, each subject in the experimental and the control groups was analyzed, and training was performed using two different sets of comorbidity features for constructing the prediction model. We proposed the novel WJI with four different machine learning techniques to construct an ALS prediction system. Since there is no diagnostic model adopting EMRs for ALS prediction, we applied both the traditional JI indicator and scoring mechanism for comparison. Furthermore, two different levels of comorbidity, individual-level disease codes and mid-level clustered disease codes, were applied for model construction and comparison of system performance.

For the ALS prediction model, the new WJI indicator-based system performed better than the traditional JI indicator and scoring mechanism-based methods. Predicting ALS disease using the WJI indicator and individual-level disease codes could yield a high accuracy of 83.7%, sensitivity of 78.8%, specificity of 85.7%, and AUC value of 0.907. When the mid-level clustered disease codes were used, the performance was slightly degraded, showing an accuracy of 80.8%, sensitivity of 72.9%, specificity of 88.6%, and AUC value of 0.844.

From these results, it can be observed that the learning models were more effective when the individual-level disease codes rather than the mid-level clustered classification were used. Thus, the WJI is more effective for model construction based on individual-level disease codes. This is because the mid-level clustered disease codes can dilute the weighted relationship and can reduce the effectiveness of the comorbidity features. The difference between the WJI and JI is that the former enhances the uneven relationship of comorbidities. When a large proportion of subjects in both the experimental and control groups suffer from an identical disease, the relative weighting coefficients and degrees of influence increase. In contrast, when the number of disease codes for various diseases is large but only a small number of subjects suffer from diseases under the same codes, the relative weighting coefficient and corresponding impact are reduced. The traditional JI similarity indicator is based on either the presence or absence of a disease code. It does not consider the different proportions and prevalence of the selected comorbidities. As only the binary conditions of 0 and 1 can be applied to evaluate comorbidities, the value of WJI could yield better distinguishability than the JI indicator.

## 4. Conclusions

In this study, the original feature sets without discriminative attributes can be improved by the novel proposed indicator and the newly modified feature sets can be trained effectively to realize a good prediction system. Although the data size is small in this study, the prediction performance with accuracy rates higher than 80% was comparable to traditional neuroimaging-based approaches. The proposed ALS prediction model is a time-saving and convenient noninvasive way to detect and evaluate at-risk ALS subjects. However, many mimicking diseases holding similar symptoms are likely to cause similar historical EMRs, and this would increase the false positive rate in general. Nevertheless, early alerts for physicians to identify at-risk ALS patients is one of the main goals of this study, and the proposed WJI indicators can be applied to construct a prediction model for a defined disease with specific comorbidities. Based on the prediction results of the personal disease records, detecting and treating potential patients in the early stages can be achieved. Therefore, the model can strengthen the prevention of specific diseases, can prolong survival years, and can improve the quality of life. It can be used to obtain an efficient in silico analytical tool to effectively aid medical applications for detecting or evaluating difficult diseases. By exploring a large number of medical records to improve the preventive medical applications, we hope that the proposed method can enable doctors

to make good decisions in terms of medical treatment and risk assessment for precise diagnosis in the future.

**Supplementary Materials:** The following are available online at https://www.mdpi.com/2076-3 417/11/3/1289/s1. Five different examples are illustrated to show the conception of the novel proposed weighted Jaccard index.

**Author Contributions:** Conceptualization, C.-H.H. and T.-W.P.; methodology, C.-H.H. and T.-W.P.; software, C.-H.H.; validation, B.-S.Y., D.T. and C.-S.H.; data curation, B.-S.Y. and C.-S.H.; writing—original draft preparation, C.-H.H.; writing—review and editing, T.-W.P.; supervision, B.-S.Y., D.T. and C.-S.H. All authors have read and agreed to the published version of the manuscript.

**Funding:** The work was supported by Ministry of Science and Technology, Taiwan (MOST 104-2321-B-019-009 to Tun-Wen Pai) and National Taipei University of Technology International Joint Research Project (NTUT-IJRP-109-08).

**Institutional Review Board Statement:** The study was conducted according to the guidelines of the Declaration of Helsinki, and approved by the Institutional Review Board of Chang Gung Memorial Hospital, Taiwan (protocol code 104-5430B and date of approval 2015/08/20). The IRB approved a request to waive the documentation of informed consent.

**Data Availability Statement:** Data sharing not applicable.

**Conflicts of Interest:** The authors declare that they have no competing interests.

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
