# Peer review of "Comorbidity Pattern Analysis for Predicting Amyotrophic Lateral Sclerosis"

_applsci, doi:10.3390/app11031289_

Round 1
Reviewer 1 Report
This paper is well written and presents a relevant topic that can add to the literature about innovative diagnostics Tools based on machine Learning methods. Abstract is well written with good balance between the different sections. First sentance is unclear, what do you mean by medical service and how is clinical trial analysis a medical service? Would recommend replacing "...can aid various medical services..." to something else, e.g. "...be useful for..." Introduction: References to support many of the statements made are largely lacking throughout the whole section, which gives the reader the impression that this is based on authors opinion rather than supported by facts and previous research (e.g. first sentance and first paragaraph...). In addition, there is no clear rationale for why the authors selected this disease for the study or how this relates to other research in this field and the potential caveats of using EMR data for this purpose (e.g. quality, completeness, structure...). Methods: Selection of comorbidities were based on crude ORs which disregards the correlations between disases. It would be informative for the reader to understand how this may have impacted the models and the results. Why did authors not consider a multivariate log reg instead to define the associations between comorbidities and ALS? That may have improved prediction and reduce noise in the ML algorithms. LL130: Would use association rather than "causing" here since the OR will not prove causality. LL133-134: I do not understand this, would recommend to describe this in another way, e.g. "...the comorbidities were selected based on the OR for comorbidities in experimental (ALS) group...." How can the number of subjects be the same in both groups? Do you rather mean number of comorbidities? Some text in the methods section should be moved to introduction, e.g. lies 256-271. Why was there a restriction in age to 49-51 years to define the disease group rather than taking all with a ALS diagnostic code (LL281)? This does no make sense to me as it will miss many cases and represent a huge underestimation of the disease, hence the prediciton models could be based on the comorbidities of only the average patient rather than the complete patient population and therefore less applicable in the general population. Results: Should be rewritten. There is a mix of methods and discussions in the results section, e.g. LL308-313 should be moved to discussion and LL315-378 to methods. Discussion - There is a lack of discussion about how these results compares to other similar studies/models and pros and cons with this approach. It would be helpful for the reader to understand this work better if the authors could elaborate more about prons and cons with their approach compared to other models and what the limitations of this work might be. There is a lack of discussion of these results and there are no references to other literature. (Minor: Line numbers were dropped in discussion and conclusions) Conclusion - too long and some this text should be moved to discussion.
Author Response
We thank reviewer's comments. All the suggestions were replied point-by-point as the following.
Reviewer 1:
This paper is well written and presents a relevant topic that can add to the literature about innovative diagnostics Tools based on machine Learning methods.
Response:
We thank reviewer’s comments.
Abstract is well written with good balance between the different sections. First sentance is unclear, what do you mean by medical service and how is clinical trial analysis a medical service? Would recommend replacing "...can aid various medical services..." to something else, e.g. "...be useful for..."
Response:
We thank reviewer’s comments. We have modified the contents according to the comments. (in Abstract)
Introduction: References to support many of the statements made are largely lacking throughout the whole section, which gives the reader the impression that this is based on authors opinion rather than supported by facts and previous research (e.g. first sentance and first paragaraph...). In addition, there is no clear rationale for why the authors selected this disease for the study or how this relates to other research in this field and the potential caveats of using EMR data for this purpose (e.g. quality, completeness, structure...).
Response:
Thank reviewer’s comments. We have removed the first paragraph from the Introduction, and we added the important messages and dilemmas of early detection of Amyotrophic lateral sclerosis (ALS) in the Introduction. The structure of the manuscript was reconstructed. (in Introduction Section)
Methods: Selection of comorbidities were based on crude ORs which disregards the correlations between diseases. It would be informative for the reader to understand how this may have impacted the models and the results.
Response:
Thank reviewer’s comments. We have added the contents to describe why we used the comorbidities to construct the prediction models (in Introduction Section), and the reason for applying odds ratio analysis (in Materials and Methods Section).
(Introduction) “……Amyotrophic lateral sclerosis (ALS) is a multi-syndrome and fatal neurodegenerative disorder that results in progressive loss of bulbar and limb function. The prevalence of ALS is uniformly distributed across most countries. According to the ALS Association, the incidence rate of ALS is approximately 1.8 to 2 per 100,000 people annually. Death from ALS occurs from respiratory failure, generally within 2–3 years from the onset of the bulbar symptom and 3–5 years after the limb onset. For the majority of patients with ALS, it usually takes 9 to 15 months from symptom onset to definitive diagnosis by two or three specialists. Currently, the etiology of sporadic ALS is unclear, and there is no cure for ALS. Treatment is mainly aimed at delaying the progression of the disease, not eliminating the symptoms. Hence, if the at-risk candidates of ALS can be identified during early stages, they can be advised to focus on strengthening their immune system and improving the environmental conditions they live in. Early prognosis may slow the progressive deterioration of ALS symptoms. Since there is a near two-year lead-time interval prior to definitive diagnosis of ALS, symptoms and comorbidities could provide important clues for ALS prediction. Hence, we consider Electronic Medical Records (EMRs) as useful features for discriminating ALS and non-ALS patients in the early stages, and the developed classifier could serve as an alternative diagnostic approach or an early warning signal for doctors and ALS patients……”
(Materials and Methods) “……For the disease code sets from both the experimental and control groups, the same lead-time interval and an identical minimum support threshold setting were applied to define the associated disease records for comorbidity analysis. To discover highly associated comorbidities with ALS, the classification codes of all the comorbidities were further analyzed by evaluating the corresponding odds ratios (ORs) for ALS. In other words, the OR of each comorbidity was calculated to evaluate the chances of a comorbidity associated with ALS. A set of comorbidity codes with ORs > 2 was constructed and defined as a significant comorbidity feature set for ALS. Since the significant comorbidity set of ALS was retrieved from the disease set of the original experimental group, after the OR analysis, the resulting association comorbidities must be a subset from the disease set of the experimental group. Therefore, the number of subjects for the filtered associated diseases in the significant comorbidity feature set should be the same as that of the subjects within the original experimental group. This process may remove background noise (unsignificant comorbidities) from the constructed prediction model. For example, the disease code sets of both the experimental and control groups may simultaneously contain a seasonal cold; hence, a small OR could be obtained and no specific correlation was detected between seasonal cold and ALS. Thus, following the statistical significance analysis of all comorbidities, the system defined a comorbidity feature set with significant association with ALS, which can explain the strong relationship between the comorbidity codes and ALS.”
Why did authors not consider a multivariate log reg instead to define the associations between comorbidities and ALS? That may have improved prediction and reduce noise in the ML algorithms.
Response:
Thank reviewer’s comments. We did apply multivariate logistic regression approaches at the beginning. However, the prediction results were not satisfied after a lot of various feature selection and feature combinations (accuracies were always near or less than 0.6). So we came up the idea of using Jaccard Index and weighted Jaccard index as indicators, and the results could be significantly improved.
LL130: Would use association rather than "causing" here since the OR will not prove causality.
Response:
We thank reviewer’s comment. We have modified the contents according to the comment.
LL133-134: I do not understand this, would recommend to describe this in another way, e.g. "...the comorbidities were selected based on the OR for comorbidities in experimental (ALS) group...." How can the number of subjects be the same in both groups? Do you rather mean number of comorbidities?
Response:
We thank reviewer’s comment. We have revised the contents to make it more clear to readers.
“….Since the significant comorbidity set of ALS was retrieved from the disease set of the original experimental group, after the OR analysis, the resulting association comorbidities must be a subset from the disease set of the experimental group. Therefore, the number of subjects for the filtered associated diseases in the significant comorbidity feature set should be the same as that of the subjects within the original experimental group. …” Hence, the number of subjects imply the number of patients not comorbidities.
Some text in the methods section should be moved to introduction, e.g. lines 256-271.
Response:
We thank reviewer’s comment. We have modified the contents according to the comment.
Why was there a restriction in age to 49-51 years to define the disease group rather than taking all with a ALS diagnostic code (LL281)? This does no make sense to me as it will miss many cases and represent a huge underestimation of the disease, hence the prediciton models could be based on the comorbidities of only the average patient rather than the complete patient population and therefore less applicable in the general population.
Response:
We thank reviewer’s comments. According to the governmental database in Taiwan, the majority onset age of ALS is near 51 years old. However, the age of subjects in the control group ranges from 3 to 82. But most subject age ranged between 45 and 65. To avoid no suitable medical records for comparison, we selected control group only based on gender proportion(compared to ALS patients). We only make sure the control group subjects were older than 55 years old without ALS in all life records. Since the average ages for the experimental groups are 51.4 years old for male subjects and 53.8 years old for female subjects, we jus want to make sure the control group subjects possessing two-year records before 51 years old for similarity measurement of comorbitity. However, the prediction model could be applied to general population since we only focus on the symptoms and comorbidities according to the significant features we observed. Therefore, the historical medical records between 49 and 51 years old were retrieved for considering as control group subjects.
Results: Should be rewritten. There is a mix of methods and discussions in the results section, e.g. LL308-313 should be moved to discussion and LL315-378 to methods.
Response:
Thanks reviewer’s comments. However, we believed that the LL308-313 and LL315-378 were not correct line numbers with respect to our submitted manuscript. Such as LL308-313 were not in the Results Section, and the contexts of LL308-313 described the threshold settings of ORs for comorbidity feature selection and supporting number of subjects for considering representative criterion. We prefer to remain LL308-313 in the Method section. Similarly, the LL315-378 were not all belonged to the Result Section either. However, we assumed that reviewer suggested the definition of ”Model Evaluation Index” to be defined in the Method Section, so we have done the revision. We would appreciate and love to revise our manuscript according to reviewer’s suggestions once the line number could be re-confirmed again.
Discussion - There is a lack of discussion about how these results compares to other similar studies/models and pros and cons with this approach. It would be helpful for the reader to understand this work better if the authors could elaborate more about prons and cons with their approach compared to other models and what the limitations of this work might be. There is a lack of discussion of these results and there are no references to other literature. (Minor: Line numbers were dropped in discussion and conclusions)
Response:
Thanks reviewer’s comments. We have added discussion to describe pros and cons of our proposed method compared to other approaches.
“…According to our survey results, our proposed machine learning method using WJI indicators is the first approach based on electronic medical records for ALS prediction. Previous ALS diagnosis mainly focused on clinical biomarkers, biological biomarkers, genetic/proteomics biomarkers, and neuroimaging indicators [Grollemund 2019]. Several machine learning techniques have been extensively applied to assist ALS diagnosis from the mentioned biomarkers. Among all different biomarker usages, using neuroimaging approach achieved the best performance of higher than 80% sensitivity in general [Muller et al, 2016, Bede, 2017]. However, these diagnostic approaches often occurred after serious symptoms onset and lead to diagnostic delay. How to apply noninvasive approaches with outstanding sensitivities and specificities at early stage has become an important issue for ALS patients, and such a diagnostic model could help at-risk ALS patients to be recruited into earlier clinical trials. It is an obvious advantage of our proposed method using individual EMRs as features to compare with comorbidity patterns of ALS patients. Though the data size is small in this study, the prediction performance is comparable to the neuroimaging based approaches. In addition, our proposed method based on historical medical records is a relatively simple, straight-forward, effective and efficient approach. Most importantly, there is no additional cost for creating any new specific data type for evaluation. ….”
Conclusion - too long and some this text should be moved to discussion.
Response:
Thanks reviewer’s comments. Some texts have been removed.

Reviewer 2 Report
The paper aims at improving the prediction of Amyotrophic Lateral Sclerosis (ALS) on base of comorbidities. Improvement is due to weighing comorbidities by their frequencies of occurrence.
However, the concept of weighed Jaccard coefficients does not alone constitute significant novelty, while the novelty of other algorithm components and the practical significance of the results of the paper cannot be deduced because of unclear writing.
The paper does not explicitly formulate the machine learning task; as the result, it is not clear how the Weighed Jaccard Indices are used in machine learning models, and furthermore, how patient scores, presented in tables 7-10 once for each classifier, are connected with those classifiers.
The introduction cites a few papers devoted to analyzing comorbidities, but does not give some insight into the methods that exist for predicting diseases by comorbidities; nor does it describe how ALS is currently diagnosed and with what accuracy. This does not allow one to compare the approach proposed to the state of the art.
In addition, a few more details lack explanation:
- what differentiation means in tables 3 to 6;
- how data is split into training and testing sets;
- how gender ratio is used when selecting the ALS-negative group.
The text needs revision, the most notable presentation flaws are the following:
- The notation for the sets (that ALS_EG and ALS_CG are groups of patients, while ALS_EG*, ALS_CG*, and ALS_FS are sets of comorbidities) is non-intuitive; it clutters the text but has little use, partly because we never see equation (1) applied to any of these sets.
- Patients with and without the disease are called group 1 and group 2 in Section 2.4, but experimental group and control group everywhere else in the paper.
- The structuring of the paper seems haphazard, especially sections 1.1 "Electronic Medical Records" and 2.6 "Amyotrophic Lateral Sclerosis (ALS)". The description of the dataset is scattered across the whole paper, appearing partially in sections 2.1 "Source of Materials", 2.6 "Amyotrophic Lateral Sclerosis (ALS)", and 3 "Results".
- Figures 1 and 2, which illustrate how patients are described by binary vectors of their comorbidities, have essentially the same structure for both groups, EG and AG. Because of that, large portions of figure captions are copied verbatim from figure 1 to figure 2, which looks unfriendly to the reader.
Due to the above, the significance of the paper can only be assessed after a major revision.
Author Response
We thank reviewer’s comments. All the suggestions were replied point-by-point as the followings.
Reviewer 2:
The paper aims at improving the prediction of Amyotrophic Lateral Sclerosis (ALS) on base of comorbidities. Improvement is due to weighing comorbidities by their frequencies of occurrence.
However, the concept of weighed Jaccard coefficients does not alone constitute significant novelty, while the novelty of other algorithm components and the practical significance of the results of the paper cannot be deduced because of unclear writing.
Response:
We thank reviewer’s comments. We have done our best to revise the manuscript for better understanding.
The paper does not explicitly formulate the machine learning task; as the result, it is not clear how the Weighed Jaccard Indices are used in machine learning models, and furthermore, how patient scores, presented in tables 7-10 once for each classifier, are connected with those classifiers.
Response:
Thanks reviewer’s comments. We have modified the method section and added more contents to describe our intents.
“….The proposed diagnostic model is mainly based on the similarity measurement of comorbidity patterns between testing subjects and known ALS patients. The similarity indicator is calculated by the proposed movel WJI similarity measurement. Once the experimental and control data groups were trained and constructed as a prediction model, the testing data subjects were evaluated according to the similarity measurement and the previously trained thresholding value. The ALS prediction module was constructed and described as the followings….”
The introduction cites a few papers devoted to analyzing comorbidities, but does not give some insight into the methods that exist for predicting diseases by comorbidities; nor does it describe how ALS is currently diagnosed and with what accuracy. This does not allow one to compare the approach proposed to the state of the art.
Response:
Thanks reviewer’s comments. We have modified the Introduction Section and reorganized the article structures. In addition, more references were cited and explained in the Introduction and Discussion Sections.
Especially, the current ALS diagnosis approaches (biomarkers) based on machine learning techniques were described and compared to our proposed method.
“…According to our survey results, our proposed machine learning method using WJI indicators is the first approach based on electronic medical records for ALS prediction. Previous ALS diagnosis mainly focused on clinical biomarkers, biological biomarkers, genetic/proteomics biomarkers, and neuroimaging indicators [Grollemund 2019]. Several machine learning techniques have been extensively applied to assist ALS diagnosis from the mentioned biomarkers. Among all different biomarker usages, using neuroimaging approach achieved the best performance of higher than 80% sensitivity in general [Muller et al, 2016, Bede, 2017]. However, these diagnostic approaches often occurred after serious symptoms onset and lead to diagnostic delay. How to apply noninvasive approaches with outstanding sensitivities and specificities at early stage has become an important issue for ALS patients, and such a diagnostic model could help at-risk ALS patients to be recruited into earlier clinical trials. It is an obvious advantage of our proposed method using individual EMRs as features to compare with comorbidity patterns of ALS patients. Though the data size is small in this study, the prediction performance is comparable to the neuroimaging based approaches. In addition, our proposed method based on historical medical records is a relatively simple, straight-forward, effective and efficient approach. Most importantly, there is no additional cost for creating any new specific data type for evaluation. ….”
In addition, a few more details lack explanation:
- what differentiation means in tables 3 to 6;
Response:
Thanks reviewer’s comments. We have added description for explaining the differentiation factors in Table 3-6.
“….The value of JI similarity between disease set of the experimental group and control group is always larger than the JI similarity between comorbidity feature set and control group. It is due to that the comorbidity feature set removed a lot of insignificant diseases by statistical analytics. Hence, several common diseases such as seasonal flu and common cold would be removed and the similarity indicator decreased. Similarly, the similarity measurement for WJI indicator possesses the same decreasing trend. From Table 3, it can be observed that the differentiation value of JI indicator (0.22) is less than the WJI indicator (0.43). It is mainly due to each disease code in feature set was weighted in WJI indicator according to the disease population, and the difference would be enlarged due to unevenly distributed subjects. Therefore, the WJI indicator could provide a better differentiating factor than JI indicator in general….”
- how data is split into training and testing sets;
Response:
Thanks reviewer’s reminding. We have added the information in the context.
“…data were utilized as the training dataset (80% of samples), the data of the remaining subjects were applied as testing data(20% of samples).…”
- how gender ratio is used when selecting the ALS-negative group.
Response:
Thanks reviewer’s reminding. We have added the information in the context.
“……71 subjects were selected for this study (male:female = 45:26)……A total number of 399 subjects without the ALS disease were selected as the control group(male:female=249:150)…….”
The text needs revision, the most notable presentation flaws are the following:
- The notation for the sets (that ALS_EG and ALS_CG are groups of patients, while ALS_EG*, ALS_CG*, and ALS_FS are sets of comorbidities) is non-intuitive;
Response:
Thanks reviewer’s comment. We have revise the notation for a better understanding.
ALS_EG and ALS_CG keep the same; ALS_EG* and ALS_CG* are renamed as ALS_EG_FEA and ALS_CG_FEA; ALS_FS is changed to ALS_SELECT_FEA
it clutters the text but has little use, partly because we never see equation (1) applied to any of these sets.
Response:
Thanks reviewer’s comment. We think the equation (1) is important to introduce weighted Jaccard index. It was use and demonstrated in the supplementary document(examples). So we prefer to retain the equation in the context.
- Patients with and without the disease are called group 1 and group 2 in Section 2.4, but experimental group and control group everywhere else in the paper.
Response:
Thanks reviewer’s comment. We didn’t see the group 1 and group 2. However, all datasets in this manuscript were consistently used as control group and experimental group.
- The structuring of the paper seems haphazard, especially sections 1.1 "Electronic Medical Records" and 2.6 "Amyotrophic Lateral Sclerosis (ALS)". The description of the dataset is scattered across the whole paper, appearing partially in sections 2.1 "Source of Materials", 2.6 "Amyotrophic Lateral Sclerosis (ALS)", and 3 "Results".
Response:
Thanks reviewer’s comment. We have revised the manuscript and the structure was tremendously changed.
- Figures 1 and 2, which illustrate how patients are described by binary vectors of their comorbidities, have essentially the same structure for both groups, EG and AG. Because of that, large portions of figure captions are copied verbatim from figure 1 to figure 2, which looks unfriendly to the reader.
Response:
Thanks reviewer’s comments. We have combined Figure 1 and Figure 2 into Figure 1(a) and (b), and redundant texts were removed.
Due to the above, the significance of the paper can only be assessed after a major revision.

Round 2
Reviewer 1 Report
Comments to authors on revised manuscript General comments:First sentance of abstract is still unclear with regards to clinical trial analysis. While researchers have used EMRs for data collection in pragmatic trials, I am not sure it is typically used to analyse clinical trials, so the wording is a bit unclear. The language could be improved if reviewed by an epidemiologist and a native English speaker as there are several faults in how the authors present epidemiological data (e.g. LL43 incidence presented in person-years and not people) and the English language should be improved. Many statements still lack references and it would be helpful if the language was kept at a scientific level with limited subjective comments (e.g. Discussion: It is an obvious advantage...). Most of the text in the Results section is actually Discussion and there is still no discussion about limitations and bias of this approach, including the potential caveats of using EMR data for this purpose commented on earlier (e.g. quality, completeness, structure...), which makes the text unbalanced. As per author instructions, authors might consider combining results and discussion or move text to Discussion for better clarity. References to support many of the statements are still missing or put in the wrong place. While authors did describe that they considered multivariate regressions, there is nothing in the manuscript about this nor any discussion regarding the choice of approach. Conclusions are not supported by results and would propose revisions to be more in line with the manuscript and potential limitations. Specific comments:
LL54, would suggest patients or subjects instead of candidates. LL59 would erase the word features here. LL114 what do you mean by "partially" selected medical data? This is unclear and appear abritrary, please explain to the reader. LL126 What is meant by "limited number of positive subjects..."? How were they selected if this is a subgroup of the complete positive data group (ALS cases)? Do you mean "...the smaller number of..."? LL128 talks about constrianing conditions, does that mean that non-cases were matched to cases on age and gender? Please use common epidemiological language like matching if so. LL132 Please explain what types of and how those statistical analyses were done on the disease codes for both groups. It sounds like the authors grouped subjects by diagnosis codes, please explain how this approach is different from a traditional case-control analysis. If I understand correctly, the authors assessed the crude association between each comorbidity and ALS with a case-control approach using logistic regression and then selected those diseases that presented at least double risk for ALS, but the way they grouped diseases by number of subjects gives the impression that it was the number of subjects for each disease that was the independent variable, please explain. An alternative might be to use an ALS specific comorbidity index (compare eg the Charlson index). LL133 the reminaing question is what is meant by "significant" here and throughout the manuscript! In the text that follows, it appears that comorbidities were selected based on the point estimates of the regression model and not the p-values or confidence intervals, if that is the case, the authors should not use the term significance since that implies a decision based on statistical certainty and not the strengh of the association which appears to be used here. LL140 Explain "certain period of time" here, it is important for readers to understand the risk window applied. LL144 Explain what the minimum support threshold setting means. LL152 Discover is a strange term here, would suggest LL159 This is unclear and should be described better so the reader can understand what was done. I guess the way this was done was that the association between each comorbidity and ALS was estimated with OR and then authors selected those comorbidities with the strongest association (OR>2). It is clear that this had to be done on the cases, ie the group called experimental group here, but it is unclear what the authors mean with a subset of the experimental group. Was the experimental group a different group from the "positive" data group, ie those who had ALS? The sentance the resulting association..." is unclear. Would suggest a better explanation of how the groups were constructed, e.g. in a figure with a patient flow chart/decision tree. LL160 what is meant by "filtered" here and what do you mean by the same number of subjects, is it because all subjects in the experimental group were ALS cases and a complete case analysis, ie the same common N, was applied for all the log reg for all comorbidities? I do not understand the explanation of why that would remove background noise or the strong relationship in the following sentences. LL155 talks about chances of a comorbidity associated with ALS, I would suggest the authors to use epidemiological language instead and describe that as the association between each comorbidity and ALS or the comorbidities as risk factors for ALS. LL157-168 I still do not understand this description and would advice authors to revise this using epidemiological language. LL166 again, was the method really based on significance or the strenght of the associations? LL172 confusing, which are these groups? If (a) is the experimental group and (b) the controls group, what is then the experimental/control group? Presume you mean either experimental or control group, this is confusing. Would suggest a flow chart explaining these groups. LL309 The inclusion of only hospitalised cases is expected to lead to selection bias towards more severe cases and authors should reason about how that impacted the results, e.g. overestimation of model accuracy. In addition, it would be useful with a sensitivity analysis to understand how big this bias is, ie to compare with a model on all 162 ALS patients. LL322 I still do not understand the rational and impact on the model of capping the control group at ages 49-51, that appears to be a very subjective and selective way and alternative ways should be considered, e.g. mathing patients on age and gender like in traditional epidemiology. This has to be explained and I would strongly suggest to apply age and gender matching instead. LL333 Again, the question is if you based the decisions on precision or strength of association? This is inconsistent in the text. LL376-381 This new section could be better placed in either methods or discussion. LL382-400 This section is repetition of what is already described in the methods and should be moved to methods or erased as it is not part of the results of this work. LL400-onwards: Most of the text appears to be discussion and could be more clear distrinction between results and discussion, or merge into one section.
Author Response
Reviewer 1 _ round2
First sentance of abstract is still unclear with regards to clinical trial analysis. While researchers have used EMRs for data collection in pragmatic trials, I am not sure it is typically used to analyse clinical trials, so the wording is a bit unclear. The language could be improved if reviewed by an epidemiologist and a native English speaker as there are several faults in how the authors present epidemiological data (e.g. LL43 incidence presented in person-years and not people) and the English language should be improved.
Response:
Thanks for reviewer’s comments.
We did have our manuscript to be revised by native English speaker (but may not be an epidemiologist). The English language editing service was done by Editage (www.editage.com). Anyway, we have modified our abstract and contents according to reviewer’s suggestions.
- Abstract: the first sentence was modified as the following.
“Electronic Medical Records (EMRs) can be used to create alerts for clinicians to discover patients at-risk and provide useful information for clinical decision-making support.”
- LL43: the incidence of ALS was re-written according to epidemiological terminology. The sentence was modified as the followings.
“According to the ALS Association, the incidence rate of ALS is approximately between 1.8 and 2 per 100,000 person-years.”
Many statements still lack references and it would be helpful if the language was kept at a scientific level with limited subjective comments (e.g. Discussion: It is an obvious advantage...). Most of the text in the Results section is actually Discussion and there is still no discussion about limitations and bias of this approach, including the potential caveats of using EMR data for this purpose commented on earlier (e.g. quality, completeness, structure...), which makes the text unbalanced. As per author instructions, authors might consider combining results and discussion or move text to Discussion for better clarity.
Response:
Thanks for valuable suggestions, we have removed subjective phrases and combined “Results” and “Discussion” sections. Limitations and bias of our approach was described as well.
- Discussion:
“… One advantage of our proposed method is that using individual EMRs as features to compare with comorbidity patterns of ALS patients is a noninvasive, cost-free, and efficient approach for early detection. However, there is no single test could provide a definitive diagnosis of ALS so far. Too many mimicking diseases hold similar symptoms and likely result in similar historical EMRs. Hence, the proposed approach could only provide an early alert for physicians. Complete neurologic examinations for muscle weakness, spasticity, and atrophy are still required for precision diagnosis of ALS. …”
References to support many of the statements are still missing or put in the wrong place. While authors did describe that they considered multivariate regressions, there is nothing in the manuscript about this nor any discussion regarding the choice of approach.
Response:
Thanks for reviewer’s suggestions. we appreciate if reviewer can point out the missing references and/or references in wrong place. We did try several different approaches for this study. However, due to the main thematic issue of novel proposed WJI indicator for this manuscript, we prefer not to raise all tried approaches. It would cause our work misfocused.
Conclusions are not supported by results and would propose revisions to be more in line with the manuscript and potential limitations.
Response:
Thanks for reviewer’s suggestions. We added results in Conclusion Section to support the conclusions. The added contents as the followings.
“…. In this study, the original feature sets without discriminative attributes can be improved by the novel proposed indicator, and the newly modified feature sets can be trained effectively to realize a good prediction system. Although the data size is small in this study, the prediction performance with accuracy rates higher than 80% are comparable to the traditional neuroimaging based approaches. The proposed ALS prediction model is a time-saving and convenient non-invasive way to detect and evaluate at-risk ALS subjects. However, many mimicking diseases holding similar symptoms are likely to cause similar historical EMRs, and this would increase false positive rate in general. Nevertheless, early alert for physicians to identify ALS patients at-risk is the main goal of this study. The proposed WJI indicators can be effectively extended to predict any other disease. Based on the prediction results of the personal disease records, detecting and treating potential patients in the early stages can be achieved…..”
LL54, would suggest patients or subjects instead of candidates.
Response:
Thanks for reviewer’s suggestions. We have modified all “candidates” to “subjects” (three words changed)
LL59 would erase the word features here.
Response:
Thanks for reviewer’s suggestions. We have removed the word “features”.
LL114 what do you mean by "partially" selected medical data? This is unclear and appear abritrary, please explain to the reader.
Response:
Thanks for reviewer’s comments. In Taiwan, the population are 23 millions. The authorized data set contains complete EMRs of one million people only. To make it clear, we rewrote the sentence as “…. The data used in this study were anonymous medical data authorized by the NHIRD (Taiwan) ; it consisted one million insured people collected between 1996 and 2013 (IRB: 104-543). ….”
LL126 What is meant by "limited number of positive subjects..."? How were they selected if this is a subgroup of the complete positive data group (ALS cases)? Do you mean "...the smaller number of..."?
Response:
Thanks for reviewer’s comments. We have rewritten the sentence to make it clearer. The “limited number” was removed. In fact, the data was retrieved from one-million population dataset instead of 23-million completed data set. Hence, we use the word ” limited” in the manuscript. Since it might cause confusion, we could simply remove the word “limited”.
“……….. The positive and negative data groups were composed of subjects with and without the ALS disease, respectively. The positive subjects were obtained directly from the one million NHIRD medical database,…”
LL128 talks about constrianing conditions, does that mean that non-cases were matched to cases on age and gender? Please use common epidemiological language like matching if so.
Response:
Yes, we have modified the contents.
“…and the relatively larger number of negative subjects were retrieved from the same database based on matching gender and age attributes according to the positive subjects….”
LL132 Please explain what types of and how those statistical analyses were done on the disease codes for both groups.
Response:
Since this is retrospective study, we used odds ratio to evaluated each disease code and described in Section 2.2.
“…the classification codes of all the comorbidities were further analyzed by evaluating the corresponding odds ratios (ORs) for ALS….”
It sounds like the authors grouped subjects by diagnosis codes, please explain how this approach is different from a traditional case-control analysis. If I understand correctly, the authors assessed the crude association between each comorbidity and ALS with a case-control approach using logistic regression and then selected those diseases that presented at least double risk for ALS, but the way they grouped diseases by number of subjects gives the impression that it was the number of subjects for each disease that was the independent variable, please explain. An alternative might be to use an ALS specific comorbidity index (compare eg the Charlson index).
Response:
Nope, we didn’t group subjects by diagnosis codes. There are only two groups of subjects including with ALS and without ALS. We only analyzed the representative/significant comorbidities and associated proportion of ALS patient to construction the prediction models.
LL133 the reminaing question is what is meant by "significant" here and throughout the manuscript! In the text that follows, it appears that comorbidities were selected based on the point estimates of the regression model and not the p-values or confidence intervals, if that is the case, the authors should not use the term significance since that implies a decision based on statistical certainty and not the strength of the association which appears to be used here.
Response:
Thanks for reviewer’s comments. As previous response, we just used odds ratio to measure the association between each comorbidity and ALS disease, and we have changed the word “significant comorbidity” to “associated comorbidity” in the manuscript. In addition, “significant association” was changed to “strong association”
LL140 Explain "certain period of time" here, it is important for readers to understand the risk window applied.
Response:
Thanks for reviewer’s comments. We have clearly described the time interval is two years in the context.
LL144 Explain what the minimum support threshold setting means.
Response:
Thanks for reviewer’s comments. We have added the definition of minimum support in the context.
“…The total number of subjects suffering from a specific disease in the experimental or control group was noted, and the comorbidity codes were preliminarily screened by evaluating a minimum support threshold setting. A minimum support threshold value represents the minimum number or percentage of the number of total subjects within the experimental group.
…”
LL152 Discover is a strange term here, would suggest
Response:
We don’t understand the suggestion from reviewer (unfinished phrase). However, “discover” seems a common word for us to use. If reviewer doesn’t like it, we could change it to “identify”. Three “discover” were modified in the manuscript.
LL159 This is unclear and should be described better so the reader can understand what was done. I guess the way this was done was that the association between each comorbidity and ALS was estimated with OR and then authors selected those comorbidities with the strongest association (OR>2). It is clear that this had to be done on the cases, ie the group called experimental group here, but it is unclear what the authors mean with a subset of the experimental group. Was the experimental group a different group from the "positive" data group, ie those who had ALS? The sentance the resulting association..." is unclear. Would suggest a better explanation of how the groups were constructed, e.g. in a figure with a patient flow chart/decision tree.
Response:
Thanks for reviewer’s comments. We are happy that the reviewer agreed with us regarding the strong association between ALS and certain diseases. To strengthen the association, the odds ratios were considered to retrieve comorbidities with stronger association. Hence these selected stronger comorbidity set was constructed as a subset from the disease set of the experimental group. The experimental group is the “positive” data group. “…the resulting association comorbidities..." was modified as “…the selected comorbidities...".
LL160 what is meant by "filtered" here and what do you mean by the same number of subjects, is it because all subjects in the experimental group were ALS cases and a complete case analysis, ie the same common N, was applied for all the log reg for all comorbidities? I do not understand the explanation of why that would remove background noise or the strong relationship in the following sentences.
Response:
Yes, after comorbidity disease screening processes, the number of subjects (positive data) keeps the same as the original positive data set. Only partial comorbidities left for constructing prediction model.
LL155 talks about chances of a comorbidity associated with ALS, I would suggest the authors to use epidemiological language instead and describe that as the association between each comorbidity and ALS or the comorbidities as risk factors for ALS.
Response:
Thanks for reviewer’s comments. But we are sorry that we are computer scientists not epidemiologists. We tried to do our best to explain what we have done for this study.
LL157-168 I still do not understand this description and would advice authors to revise this using epidemiological language.
Response:
Thanks for reviewer’s comments. To tell the truth, we all believe that an epidemiological expert is not easy to be trained within a short period of time. Hopefully, I think the majority readers of Applied Science are not epidemiologists. However, iff the reviewer can provide more valuable advices from the epidemiologic aspects, we would love to learn and revise our manuscript in that way.
LL166 again, was the method really based on significance or the strength of the associations?
Response:
Thanks for reviewer’s comments. The odds ratio is applied to measure association for a case-control study. We applied odds ratios to evaluate the strength of association between comorbidities and ALS disease. We have modified all the word “significant” into “associated” through the manuscript.
LL172 confusing, which are these groups? If (a) is the experimental group and (b) the controls group, what is then the experimental/control group? Presume you mean either experimental or control group, this is confusing. Would suggest a flow chart explaining these groups.
Response:
Thanks for reviewer’s comments. In engineering sense, “for experimental/control groups” represents “for experimental group and for control group respectively”. We already modified the description in the combined Figure 1. (Actually, the second reviewer suggested us to combine the first two figures to avoid repeating descriptions in the first round review).
LL309 The inclusion of only hospitalised cases is expected to lead to selection bias towards more severe cases and authors should reason about how that impacted the results, e.g. overestimation of model accuracy. In addition, it would be useful with a sensitivity analysis to understand how big this bias is, ie to compare with a model on all 162 ALS patients.
Response:
Thanks for reviewer’s comments. This excluding processes belonged to data cleaning procedures. Since there are too many mimicking diseases holding similar symptoms as ALS patients, sometimes hypothetical diagnosis of ALS by doctors would occur frequently. Hence, using un-hospitalized ALS cases from EMRs would run a risk for constructing a wrong prediction model. We insist to use those who were hospitalized due to ALS disease as our positive dataset and their corresponding comorbidity records within previous two years are considered as true effective comorbidities.
LL322 I still do not understand the rational and impact on the model of capping the control group at ages 49-51, that appears to be a very subjective and selective way and alternative ways should be considered, e.g. matching patients on age and gender like in traditional epidemiology. This has to be explained and I would strongly suggest to apply age and gender matching instead.
Response:
Thanks for reviewer’s comments. We have described in the manuscript that the average onset age of the ALS subjects in Taiwan is 51 years old. We would like to know the most popular symptoms of ALS patients (at middle age) instead of other chronic diseases or genetic diseases. Hence, we decided to matching patients on gender only in this study. For example, we have a very young age ALS patient(4 years old, it might occur due to genetic heritage conditions. If we matched with a healthy kid of the same age, and his/her previous EMRs would not gain any valuable information. However, the young ALS patient still possessed similar progressive ALS symptoms as those ALS patients at middle age. Hence, we took subjects in control group at age 49-51.
LL333 Again, the question is if you based the decisions on precision or strength of association? This is inconsistent in the text.
Response:
Thanks for reviewer’s comments. Our manuscript is based on strength association through odds ratio analysis. We should avoid using statistically significant in the manuscript.
LL376-381 This new section could be better placed in either methods or discussion.
Response:
Thanks for reviewer’s comments. We have moved this newly added paragraph to the “discussion” part, and the Result and Discussion sections were merged as one section.
LL382-400 This section is repetition of what is already described in the methods and should be moved to methods or erased as it is not part of the results of this work.
Response:
Thanks for reviewer’s comments. We have removed the repeated contexts according to reviewer’s suggestions.
LL400-onwards: Most of the text appears to be discussion and could be more clear distrinction between results and discussion, or merge into one section.
Response:
Thanks for reviewer’s comments. We have merged these two sections into one section.

Reviewer 2 Report
In this revision of the paper, literature survey has been extended, and the methodological part has been somewhat clarified. However, the lack of explicit formulation of the machine learning task still has not been addressed: it is unclear what the input and output os for the classifier listed In Section 2.3. The paper uses the wording "experimental and control data groups were trained and constructed as a prediction model" in this regard, which makes very little sense. More generally, the paper might benefit from a scheme (either verbose or in the form of a diagram) that would depict, in what order different components of the proposed algorithm (selecting the most significant comorbidities; computing the Weighed Jaccard Indices; computing similarities; choosing a threshold for these similarities; training classifiers from Section 2.3) are applied. Section "Proposed Patient Prediction Module" is, I believe, quite an appropriate place for such a scheme.
Without such clarification, I am still struggling to understand and assess the algorithm proposed.
Author Response
In this revision of the paper, literature survey has been extended, and the methodological part has been somewhat clarified. However, the lack of explicit formulation of the machine learning task still has not been addressed: it is unclear what the input and output os for the classifier listed In Section 2.3. The paper uses the wording "experimental and control data groups were trained and constructed as a prediction model" in this regard, which makes very little sense. More generally, the paper might benefit from a scheme (either verbose or in the form of a diagram) that would depict, in what order different components of the proposed algorithm (selecting the most significant comorbidities; computing the Weighed Jaccard Indices; computing similarities; choosing a threshold for these similarities; training classifiers from Section 2.3) are applied. Section "Proposed Patient Prediction Module" is, I believe, quite an appropriate place for such a scheme.
Without such clarification, I am still struggling to understand and assess the algorithm proposed.
Response:
We thank reviewer’s comments. We have added one new section (Section 2.3) to describe the proposed ML scheme in details. Since we only used traditional ML techniques, there is no need to draw a figure in the manuscript. In addition, the selected comorbidities with strong association were provided as the second supplementary document.
“The proposed diagnostic model is mainly based on the similarity measurement of comorbidity patterns between testing subjects and known ALS patients. From experimental and control data groups, we initially calculated the odds ratios for each comorbidity (within a two-year interval) for measuring the association between comorbidities and ALS disease. Based on odds ratio analysis, comorbidities with strong associations were selected as important features, and the proposed WJI similarity measurement were further calculated between experimental and control data groups, and the WJI indicator was trained to develop a prediction model for binary outcomes that could support clinical decision making. Once the experimental and control data groups were trained and constructed as a prediction model, only the selected associated comorbidities of testing subjects were considered and evaluated according to the WJI similarity measurement and compared to the previously trained thresholding value. The ALS prediction module was constructed and described as the followings. Obtaining the comorbidities for each subject in the experimental or control group, and it’s corresponding WJI similarity was calculated with respect to the defined associated comorbidity feature set. Subsequently, a normalization procedure was performed for similarity measurement and for training and construction of the probability prediction model. Excluding the subjects in the experimental and control data groups, whose data were utilized as the training dataset (80% of samples), the data of the remaining subjects were applied as testing data(20% of samples). “
In the Result section, we have added the second supplementary document.
“…We have analyzed comorbidities of ALS disease with strong association through odds ratio analysis. The selected comorbidities were categorized into individual level and mid-level respectively. Both selected comorbidities with strong associations in different levels were shown in the supplementary document II….”

Round 3
Reviewer 1 Report
Comments on the resolutions of the OLD COMMENTS
(2nd comments in black, 2nd author response in blue, 3nd comments in red).
1. First sentance of abstract is still unclear with regards to clinical trial analysis. While researchers have used EMRs for data collection in pragmatic trials, I am not sure it is typically used to analyse clinical trials, so the wording is a bit unclear. The language could be improved if reviewed by an epidemiologist and a native English speaker as there are several faults in how the authors present epidemiological data (e.g. LL43 incidence presented in person-years and not people) and the English language should be improved.
Response: Thanks for reviewer’s comments. We did have our manuscript to be revised by native English speaker (but may not be an epidemiologist). The English language editing service was done by Editage (www.editage.com). Anyway, we have modified our abstract and contents according to reviewer’s suggestions. Abstract: the first sentence was modified as the following. “Electronic Medical Records (EMRs) can be used to create alerts for clinicians to discover patients at-risk and provide useful information for clinical decision-making support.” LL43: the incidence of ALS was re-written according to epidemiological terminology. The sentence was modified as the followings. “According to the ALS Association, the incidence rate of ALS is approximately between 1.8 and 2 per 100,000 person-years.”
-> This is more clear now, thank you.
2.Many statements still lack references and it would be helpful if the language was kept at a scientific level with limited subjective comments (e.g. Discussion: It is an obvious advantage...). Most of the text in the Results section is actually Discussion and there is still no discussion about limitations and bias of this approach, including the potential caveats of using EMR data for this purpose commented on earlier (e.g. quality, completeness, structure...), which makes the text unbalanced. As per author instructions, authors might consider combining results and discussion or move text to Discussion for better clarity.
Response: Thanks for valuable suggestions, we have removed subjective phrases and combined “Results” and “Discussion” sections. Limitations and bias of our approach was described as well. Discussion: “… One advantage of our proposed method is that using individual EMRs as features to compare with comorbidity patterns of ALS patients is a noninvasive, cost-free, and efficient approach for early detection. However, there is no single test could provide a definitive diagnosis of ALS so far. Too many mimicking diseases hold similar symptoms and likely result in similar historical EMRs. Hence, the proposed approach could only provide an early alert for physicians. Complete neurologic examinations for muscle weakness, spasticity, and atrophy are still required for precision diagnosis of ALS. …”
-> Thank you for clarifying this. However, I do not fully understand the new explanations. What do you mean by using EMRs as features? What do you mean by "Too many mimicking diseases hold similar symptoms and likely result in similar historical EMRs."? I am not sure that diseases are mimicking each other, but patients with different diseases could possibly present similar symptoms so there might be similarities in symptoms between diseases, and I am not sure that symptoms would result in EMRs. Readers may find this sentance unclear.
3. References to support many of the statements are still missing or put in the wrong place. While authors did describe that they considered multivariate regressions, there is nothing in the manuscript about this nor any discussion regarding the choice of approach.
Response: Thanks for reviewer’s suggestions. we appreciate if reviewer can point out the missing references and/or references in wrong place. We did try several different approaches for this study. However, due to the main thematic issue of novel proposed WJI indicator for this manuscript, we prefer not to raise all tried approaches. It would cause our work misfocused.
-> This is more clear now, thank you. Examples of missing refs were: LL75 "According to research reports..." While the following paragraph includes some examples, this sentance appear out of place with no reference to support the statement. LL88 This sentance could end with reference 13 (which follows after subsequent sentances). Regarding the question about trying different models, I do not agree with the authors as I still believe it would be more transperent for reader if that was mentioned and I would recommend the authors to consider adding a statement about that in the manuscript, e.g. "Different multivariate models were also considered (tested?), but was not used in the end because of..."
4. Conclusions are not supported by results and would propose revisions to be more in line with the manuscript and potential limitations.
Response: Thanks for reviewer’s suggestions. We added results in Conclusion Section to support the conclusions. The added contents as the followings. “…. In this study, the original feature sets without discriminative attributes can be improved by the novel proposed indicator, and the newly modified feature sets can be trained effectively to realize a good prediction system. Although the data size is small in this study, the prediction performance with accuracy rates higher than 80% are comparable to the traditional neuroimaging based approaches. The proposed ALS prediction model is a time-saving and convenient non-invasive way to detect and evaluate at-risk ALS subjects. However, many mimicking diseases holding similar symptoms are likely to cause similar historical EMRs, and this would increase false positive rate in general. Nevertheless, early alert for physicians to identify ALS patients at-risk is the main goal of this study. The proposed WJI indicators can be effectively extended to predict any other disease. Based on the prediction results of the personal disease records, detecting and treating potential patients in the early stages can be achieved…..”
-> The conclusions are now more balanced and combined with the results and this reads much better now, thank you. Now it's also acknowledging some of the limitations like the small sample size. However, there are still sentances in the conclusions that are not fully supported by the results (e.g. LL529 "...can be extended to other diseases..."; LL532) and it is also unclear how the conclusions relate to the aim of the project, e.g. was the aim to detect at risk patient or to develop and test a new prediction model? I would recommend to balance the conclusions further and describe what such models could be used for in more general terms rather than to explicitly state that this particular model can prolong survival etc, which was tested in this study.
5. LL54, would suggest patients or subjects instead of candidates.
Response: Thanks for reviewer’s suggestions. We have modified all “candidates” to “subjects” (three words changed)
-> Resolved.
6. LL59 would erase the word features here.
Response: Thanks for reviewer’s suggestions. We have removed the word “features”.
-> I would recommend the authors to consider the use of "features" in other places of the manuscript for clarity, e.g. LL133; LL318 (comorbidities?), LL420 (symptoms?)
7. LL114 what do you mean by "partially" selected medical data? This is unclear and appear abritrary, please explain to the reader.
Response: Thanks for reviewer’s comments. In Taiwan, the population are 23 millions. The authorized data set contains complete EMRs of one million people only. To make it clear, we rewrote the sentence as “…. The data used in this study were anonymous medical data authorized by the NHIRD (Taiwan) ; it consisted one million insured people collected between 1996 and 2013 (IRB: 104-543). ….”
-> That is more clear now, thank you. Would recommend changing "...; it consisted..." to "...consisting of..."
8. L126 What is meant by "limited number of positive subjects..."? How were they selected if this is a subgroup of the complete positive data group (ALS cases)? Do you mean "...the smaller number of..."?
Response: Thanks for reviewer’s comments. We have rewritten the sentence to make it clearer. The “limited number” was removed. In fact, the data was retrieved from one-million population dataset instead of 23-million completed data set. Hence, we use the word ” limited” in the manuscript. Since it might cause confusion, we could simply remove the word “limited”. “……….. The positive and negative data groups were composed of subjects with and without the ALS disease, respectively. The positive subjects were obtained directly from the one million NHIRD medical database,…”
-> Resolved.
9. LL128 talks about constrianing conditions, does that mean that non-cases were matched to cases on age and gender? Please use common epidemiological language like matching if so.
Response: Yes, we have modified the contents. “…and the relatively larger number of negative subjects were retrieved from the same database based on matching gender and age attributes according to the positive subjects….”
->Resolved.
10. LL132 Please explain what types of and how those statistical analyses were done on the disease codes for both groups.
Response: Since this is retrospective study, we used odds ratio to evaluated each disease code and described in Section 2.2. “…the classification codes of all the comorbidities were further analyzed by evaluating the corresponding odds ratios (ORs) for ALS….”
-> Resolved.
11. It sounds like the authors grouped subjects by diagnosis codes, please explain how this approach is different from a traditional case-control analysis. If I understand correctly, the authors assessed the crude association between each comorbidity and ALS with a case-control approach using logistic regression and then selected those diseases that presented at least double risk for ALS, but the way they grouped diseases by number of subjects gives the impression that it was the number of subjects for each disease that was the independent variable, please explain. An alternative might be to use an ALS specific comorbidity index (compare eg the Charlson index).
Response: Nope, we didn’t group subjects by diagnosis codes. There are only two groups of subjects including with ALS and without ALS. We only analyzed the representative/significant comorbidities and associated proportion of ALS patient to construction the prediction models.
-> OK, but here you talk about significant again which is confusing since the selected comorbidities were based only on the point estimate and the magnitude of the association, not whether the associations were statistically significant (see next comment).
12. LL133 the reminaing question is what is meant by "significant" here and throughout the manuscript! In the text that follows, it appears that comorbidities were selected based on the point estimates of the regression model and not the p-values or confidence intervals, if that is the case, the authors should not use the term significance since that implies a decision based on statistical certainty and not the strength of the association which appears to be used here.
Response: Thanks for reviewer’s comments. As previous response, we just used odds ratio to measure the association between each comorbidity and ALS disease, and we have changed the word “significant comorbidity” to “associated comorbidity” in the manuscript. In addition, “significant association” was changed to “strong association”
-> I still find the sentance unclear. With "associated disease features", do you mean that features such as symptoms written as phycisian case notes were analysed for each comorbidity or do you mean that diagnostic codes were used to assess which comorbidities (disease codes) that were associated with ALS? If the latter, as I understand was applied here, then I would recommend to revise the sentance accordingly. In addition, the word significant is still used in this meaning in other places (e.g. LL158, LL378, LL388, LL511).
13. LL140 Explain "certain period of time" here, it is important for readers to understand the risk window applied.
Response: Thanks for reviewer’s comments. We have clearly described the time interval is two years in the context.
->Resolved.
14. LL144 Explain what the minimum support threshold setting means.
Response: Thanks for reviewer’s comments. We have added the definition of minimum support in the context. “…The total number of subjects suffering from a specific disease in the experimental or control group was noted, and the comorbidity codes were preliminarily screened by evaluating a minimum support threshold setting. A minimum support threshold value represents the minimum number or percentage of the number of total subjects within the experimental group…”
->I do not understand what is meant by "minimum number of total subjects" or "percentage of the total number of subjects", nor the use of "setting" here. This is still unclear and unresolved.
15. LL152 Discover is a strange term here, would suggest.
Response: We don’t understand the suggestion from reviewer (unfinished phrase). However, “discover” seems a common word for us to use. If reviewer doesn’t like it, we could change it to “identify”. Three “discover” were modified in the manuscript.
->Resolved. This is more clear now as discover was not the proper word.
16. LL159 This is unclear and should be described better so the reader can understand what was done. I guess the way this was done was that the association between each comorbidity and ALS was estimated with OR and then authors selected those comorbidities with the strongest association (OR>2). It is clear that this had to be done on the cases, ie the group called experimental group here, but it is unclear what the authors mean with a subset of the experimental group. Was the experimental group a different group from the "positive" data group, ie those who had ALS? The sentance the resulting association..." is unclear. Would suggest a better explanation of how the groups were constructed, e.g. in a figure with a patient flow chart/decision tree.
Response: Thanks for reviewer’s comments. We are happy that the reviewer agreed with us regarding the strong association between ALS and certain diseases. To strengthen the association, the odds ratios were considered to retrieve comorbidities with stronger association. Hence these selected stronger comorbidity set was constructed as a subset from the disease set of the experimental group. The experimental group is the “positive” data group. “…the resulting association comorbidities..." was modified as “…the selected comorbidities...".
->Resolved.
17. LL160 what is meant by "filtered" here and what do you mean by the same number of subjects, is it because all subjects in the experimental group were ALS cases and a complete case analysis, ie the same common N, was applied for all the log reg for all comorbidities? I do not understand the explanation of why that would remove background noise or the strong relationship in the following sentences.
Response: Yes, after comorbidity disease screening processes, the number of subjects (positive data) keeps the same as the original positive data set. Only partial comorbidities left for constructing prediction model.
->Resolved.
18. LL155 talks about chances of a comorbidity associated with ALS, I would suggest the authors to use epidemiological language instead and describe that as the association between each comorbidity and ALS or the comorbidities as risk factors for ALS.
Response: Thanks for reviewer’s comments. But we are sorry that we are computer scientists not epidemiologists. We tried to do our best to explain what we have done for this study.
->Hence the recommendation. While I appreciate the novelty and importance of data science in order to improve the use of big data in healthcare, given the comments here it appears likely that the quality of this work could be improved if traditional biostatics and epidemiological concepts were considered, e.g. in the assessment of the associations between comorbidities and ALS by accounting for bias and precision rather than just a crude assessment of point estimates, and in the intepretations and writing process.
19. LL157-168 I still do not understand this description and would advice authors to revise this using epidemiological language.
Response: Thanks for reviewer’s comments. To tell the truth, we all believe that an epidemiological expert is not easy to be trained within a short period of time. Hopefully, I think the majority readers of Applied Science are not epidemiologists. However, iff the reviewer can provide more valuable advices from the epidemiologic aspects, we would love to learn and revise our manuscript in that way.
->The recommendation was not about training but consultation. Data science is an emerging and increasingly important field that can dramatically improve the use of big data in healthcare, but consulting a biostatistician or epidemiologist about the design and interpretations of this study could improve the quality and dissemination. Examples include language such as significant, bias, person-years, etc as commented earlier.
20. LL166 again, was the method really based on significance or the strength of the associations?
Response: Thanks for reviewer’s comments. The odds ratio is applied to measure association for a case-control study. We applied odds ratios to evaluate the strength of association between comorbidities and ALS disease. We have modified all the word “significant” into “associated” through the manuscript.
->Resolved.
21. LL172 confusing, which are these groups? If (a) is the experimental group and (b) the controls group, what is then the experimental/control group? Presume you mean either experimental or control group, this is confusing. Would suggest a flow chart explaining these groups.
Response: Thanks for reviewer’s comments. In engineering sense, “for experimental/control groups” represents “for experimental group and for control group respectively”. We already modified the description in the combined Figure 1. (Actually, the second reviewer suggested us to combine the first two figures to avoid repeating descriptions in the first round review).
->Understood.
22. LL309 The inclusion of only hospitalised cases is expected to lead to selection bias towards more severe cases and authors should reason about how that impacted the results, e.g. overestimation of model accuracy. In addition, it would be useful with a sensitivity analysis to understand how big this bias is, ie to compare with a model on all 162 ALS patients.
Response: Thanks for reviewer’s comments. This excluding processes belonged to data cleaning procedures. Since there are too many mimicking diseases holding similar symptoms as ALS patients, sometimes hypothetical diagnosis of ALS by doctors would occur frequently. Hence, using un-hospitalized ALS cases from EMRs would run a risk for constructing a wrong prediction model. We insist to use those who were hospitalized due to ALS disease as our positive dataset and their corresponding comorbidity records within previous two years are considered as true effective comorbidities.
-> I understand that restriction to hospitalised cases may make sense if the intention is to construct a prediction model that is as accurate as possible, but it would strenghen the manusctipt if the authors could discuss this and any limitations in the use of the model due to that restriction. If the prediction model is based on a highly selected patient population and then applied to the broader patient population, could the model accuracy be overestimated as it was not accounting for the diversity of patients in the EMRs where it is supposed to be implemented for e.g. screening or early diagnosis in the end? Or will the prediction model work better on ALS patients with certain comorbidities, on the more severe cases, or only on hospitalised patients? Would the prediction model perform worse in detecting early ALS if the model was trained on the broader patient population (all adults including all comorbidities)? It would be interesting for the reader to understand how the authors as experts in this field view that.
23. LL322 I still do not understand the rationale and impact on the model of capping the control group at ages 49-51, that appears to be a very subjective and selective way and alternative ways should be considered, e.g. matching patients on age and gender like in traditional epidemiology. This has to be explained and I would strongly suggest to apply age and gender matching instead.
Response: Thanks for reviewer’s comments. We have described in the manuscript that the average onset age of the ALS subjects in Taiwan is 51 years old. We would like to know the most popular symptoms of ALS patients (at middle age) instead of other chronic diseases or genetic diseases. Hence, we decided to matching patients on gender only in this study. For example, we have a very young age ALS patient (4 years old, it might occur due to genetic heritage conditions. If we matched with a healthy kid of the same age, and his/her previous EMRs would not gain any valuable information. However, the young ALS patient still possessed similar progressive ALS symptoms as those ALS patients at middle age. Hence, we took subjects in control group at age 49-51.
-> I am concerned about this approach. That would mean that you disregards all other diagnosis-disease associations and create a prediction model that is only based on the associations among patients 49-51? Would that not overestimate the model accuracy and limit its application? I would like to see what the results would be based on a broader age group of eg all adults and at least a discussion about any implications of this in line with the previous comment.
24. LL333 Again, the question is if you based the decisions on precision or strength of association? This is inconsistent in the text.
Response: Thanks for reviewer’s comments. Our manuscript is based on strength association through odds ratio analysis. We should avoid using statistically significant in the manuscript.
->To avoid this issue, I would strongly recommend to search the document again, as it appears that the word is still used (e.g. LL158, LL378, LL388, LL511).
25. LL376-381 This new section could be better placed in either methods or discussion.
Response: Thanks for reviewer’s comments. We have moved this newly added paragraph to the “discussion” part, and the Result and Discussion sections were merged as one section.
->Resolved.
26. LL382-400 This section is repetition of what is already described in the methods and should be moved to methods or erased as it is not part of the results of this work.
Response: Thanks for reviewer’s comments. We have removed the repeated contexts according to reviewer’s suggestions.
->Resolved.
27. LL400-onwards: Most of the text appears to be discussion and could be more clear distrinction between results and discussion, or merge into one section. Response: Thanks for reviewer’s comments. We have merged these two sections into one section.
->Resolved.
NEW COMMENTS
LL119. What is meant by "...verify the effectiveness of the novel proposed indicator" in the aims? I cannot see that there was any test of its performance that would verify its effectiveness. For that, I would expect to see results where this so called indicator was applied an EMR system for screening ALS patients or early diagnosis. LL220 Can you really say that the evaluation is "precise"? On what grounds? Or do you mean "more precise" than the traditional JI? LL221 Did you estimate incidence rates in this study? LL486 Would recommend to describe it as "...using EMRs as the source data to compare comorbidity patterns..." LL488 Would recommend to revise this sentance as it may appear unclear to the reader (...mimicking diseases..."). LL491 Use precise instead of precision.
Author Response
Editor note:
Reviewer's 2rd comments (Black) ;
Author response for 2rd comments (Blue) ;
Reviewer's 3rd comments (Red) ;
Author response for 3rd comments (Brown) ;
-------------------------------------------------------
Thanks for reviewer’s comments. We only answered the newly proposed questions (in brown). The resolved problems were removed from this response document.
2.Many statements still lack references and it would be helpful if the language was kept at a scientific level with limited subjective comments (e.g. Discussion: It is an obvious advantage...). Most of the text in the Results section is actually Discussion and there is still no discussion about limitations and bias of this approach, including the potential caveats of using EMR data for this purpose commented on earlier (e.g. quality, completeness, structure...), which makes the text unbalanced. As per author instructions, authors might consider combining results and discussion or move text to Discussion for better clarity.
Response: Thanks for valuable suggestions, we have removed subjective phrases and combined “Results” and “Discussion” sections. Limitations and bias of our approach was described as well. Discussion: “… One advantage of our proposed method is that using individual EMRs as features to compare with comorbidity patterns of ALS patients is a noninvasive, cost-free, and efficient approach for early detection. However, there is no single test could provide a definitive diagnosis of ALS so far. Too many mimicking diseases hold similar symptoms and likely result in similar historical EMRs. Hence, the proposed approach could only provide an early alert for physicians. Complete neurologic examinations for muscle weakness, spasticity, and atrophy are still required for precision diagnosis of ALS. …”
-> Thank you for clarifying this. However, I do not fully understand the new explanations. What do you mean by using EMRs as features? What do you mean by "Too many mimicking diseases hold similar symptoms and likely result in similar historical EMRs."? I am not sure that diseases are mimicking each other, but patients with different diseases could possibly present similar symptoms so there might be similarities in symptoms between diseases, and I am not sure that symptoms would result in EMRs. Readers may find this sentance unclear.
Response: Thanks reviewer’s comments. We used the words “mimic” to describe the similar disease symptoms according to ALS society (https://alstreatment.com/diseases-that-can-mimic-als/). However, we agree with your suggestion. We have removed the word “mimic” and removed the phrase “result in EMRs”. The phrase was modified as:
“…Several diseases such as multiple sclerosis and Parkinson’s diseases hold similar symptoms as ALS, and these symptoms may be initially neglected by non-ALS trained physicians…”(LL494).
3. References to support many of the statements are still missing or put in the wrong place. While authors did describe that they considered multivariate regressions, there is nothing in the manuscript about this nor any discussion regarding the choice of approach.
Response: Thanks for reviewer’s suggestions. we appreciate if reviewer can point out the missing references and/or references in wrong place. We did try several different approaches for this study. However, due to the main thematic issue of novel proposed WJI indicator for this manuscript, we prefer not to raise all tried approaches. It would cause our work misfocused.
-> This is more clear now, thank you. Examples of missing refs were: LL75 "According to research reports..." While the following paragraph includes some examples, this sentance appear out of place with no reference to support the statement. LL88 This sentance could end with reference 13 (which follows after subsequent sentances). Regarding the question about trying different models, I do not agree with the authors as I still believe it would be more transperent for reader if that was mentioned and I would recommend the authors to consider adding a statement about that in the manuscript, e.g. "Different multivariate models were also considered (tested?), but was not used in the end because of..."
Response: Thanks for the comments, we have modified the reference locations. Besides, we added the sentences to describe why we didn’t use multivariate models in this study. The
“…In this study, ALS was applied as the target disease for validating the proposed method. As a matter of fact, different multivariate models were considered initially, but the performance was not good and not reported in this study. It is believed that too many comorbidities and less ALS subjects were the main reasons causing the unsatisfied results. Therefore, we proposed the novel similarity-based approach instead of the statistical modeling approach….” (LINE 471-475).
- Conclusions are not supported by results and would propose revisions to be more in line with the manuscript and potential limitations.
Response: Thanks for reviewer’s suggestions. We added results in Conclusion Section to support the conclusions. The added contents as the followings. “…. In this study, the original feature sets without discriminative attributes can be improved by the novel proposed indicator, and the newly modified feature sets can be trained effectively to realize a good prediction system. Although the data size is small in this study, the prediction performance with accuracy rates higher than 80% are comparable to the traditional neuroimaging based approaches. The proposed ALS prediction model is a time-saving and convenient non-invasive way to detect and evaluate at-risk ALS subjects. However, many mimicking diseases holding similar symptoms are likely to cause similar historical EMRs, and this would increase false positive rate in general. Nevertheless, early alert for physicians to identify ALS patients at-risk is the main goal of this study. The proposed WJI indicators can be effectively extended to predict any other disease. Based on the prediction results of the personal disease records, detecting and treating potential patients in the early stages can be achieved…..”
-> The conclusions are now more balanced and combined with the results and this reads much better now, thank you. Now it's also acknowledging some of the limitations like the small sample size. However, there are still sentances in the conclusions that are not fully supported by the results (e.g. LL529 "...can be extended to other diseases..."; LL532) and it is also unclear how the conclusions relate to the aim of the project, e.g. was the aim to detect at risk patient or to develop and test a new prediction model? I would recommend to balance the conclusions further and describe what such models could be used for in more general terms rather than to explicitly state that this particular model can prolong survival etc, which was tested in this study.
Response:
Thanks for the comments. The phrase was modified as “ …Nevertheless, early alert for physicians to identify ALS patients at-risk is one of the main goals of this study, and the proposed WJI indicators can be applied to construct a prediction model for a defined disease with specific comorbidities.…”
- LL59 would erase the word features here.
Response: Thanks for reviewer’s suggestions. We have removed the word “features”.
-> I would recommend the authors to consider the use of "features" in other places of the manuscript for clarity, e.g. LL133; LL318 (comorbidities?), LL420 (symptoms?)
Response: Thanks for reviewer’s comments. All modified according to reviewer’s suggestions.
- LL114 what do you mean by "partially" selected medical data? This is unclear and appear abritrary, please explain to the reader.
Response: Thanks for reviewer’s comments. In Taiwan, the population are 23 millions. The authorized data set contains complete EMRs of one million people only. To make it clear, we rewrote the sentence as “…. The data used in this study were anonymous medical data authorized by the NHIRD (Taiwan) ; it consisted one million insured people collected between 1996 and 2013 (IRB: 104-543). ….”
-> That is more clear now, thank you. Would recommend changing "...; it consisted..." to "...consisting of..."
Response: Thanks for reviewer’s comments. It was modified according to reviewer’s suggestions.
- LL133 the reminaing question is what is meant by "significant" here and throughout the manuscript! In the text that follows, it appears that comorbidities were selected based on the point estimates of the regression model and not the p-values or confidence intervals, if that is the case, the authors should not use the term significance since that implies a decision based on statistical certainty and not the strength of the association which appears to be used here.
Response: Thanks for reviewer’s comments. As previous response, we just used odds ratio to measure the association between each comorbidity and ALS disease, and we have changed the word “significant comorbidity” to “associated comorbidity” in the manuscript. In addition, “significant association” was changed to “strong association”
-> I still find the sentance unclear. With "associated disease features", do you mean that features such as symptoms written as phycisian case notes were analysed for each comorbidity or do you mean that diagnostic codes were used to assess which comorbidities (disease codes) that were associated with ALS? If the latter, as I understand was applied here, then I would recommend to revise the sentance accordingly. In addition, the word significant is still used in this meaning in other places (e.g. LL158, LL378, LL388, LL511).
Response: Yes, the diagnostic codes were used to assess which comorbidities (disease codes) that were associated with ALS. Thanks for understanding the proposed conception. All the “significant” words were modified or removed from the context.
- LL144 Explain what the minimum support threshold setting means.
Response: Thanks for reviewer’s comments. We have added the definition of minimum support in the context. “…The total number of subjects suffering from a specific disease in the experimental or control group was noted, and the comorbidity codes were preliminarily screened by evaluating a minimum support threshold setting. A minimum support threshold value represents the minimum number of subjects or the minimum percentage of total subjects within the experimental group. In other words, comorbidities which were present in less than a certain percentage of subjects within the experimental group were discarded from the feature set in this study.…”
->I do not understand what is meant by "minimum number of total subjects" or "percentage of the total number of subjects", nor the use of "setting" here. This is still unclear and unresolved.
Response: Thanks for the comments. There are basic terminologies in the data mining field. Anyway, we modified the sentence as “…A minimum support threshold value represents the minimum number of subjects or the minimum percentage of total subjects within the experimental group….”
- LL155 talks about chances of a comorbidity associated with ALS, I would suggest the authors to use epidemiological language instead and describe that as the association between each comorbidity and ALS or the comorbidities as risk factors for ALS.
Response: Thanks for reviewer’s comments. But we are sorry that we are computer scientists not epidemiologists. We tried to do our best to explain what we have done for this study.
->Hence the recommendation. While I appreciate the novelty and importance of data science in order to improve the use of big data in healthcare, given the comments here it appears likely that the quality of this work could be improved if traditional biostatics and epidemiological concepts were considered, e.g. in the assessment of the associations between comorbidities and ALS by accounting for bias and precision rather than just a crude assessment of point estimates, and in the intepretations and writing process.
Response: Thanks for the comments. We have tried our best to improve the manuscript.
- LL157-168 I still do not understand this description and would advice authors to revise this using epidemiological language.
Response: Thanks for reviewer’s comments. To tell the truth, we all believe that an epidemiological expert is not easy to be trained within a short period of time. Hopefully, I think the majority readers of Applied Science are not epidemiologists. However, iff the reviewer can provide more valuable advices from the epidemiologic aspects, we would love to learn and revise our manuscript in that way.
->The recommendation was not about training but consultation. Data science is an emerging and increasingly important field that can dramatically improve the use of big data in healthcare, but consulting a biostatistician or epidemiologist about the design and interpretations of this study could improve the quality and dissemination. Examples include language such as significant, bias, person-years, etc as commented earlier.
Response: Thanks for the comments. We did learn a lot from reviewers. We also appreciate your reviewing task. It is clearly that the manuscript was improved according to reviewer’s suggestions.
- LL309 The inclusion of only hospitalised cases is expected to lead to selection bias towards more severe cases and authors should reason about how that impacted the results, e.g. overestimation of model accuracy. In addition, it would be useful with a sensitivity analysis to understand how big this bias is, ie to compare with a model on all 162 ALS patients.
Response: Thanks for reviewer’s comments. This excluding processes belonged to data cleaning procedures. Since there are too many mimicking diseases holding similar symptoms as ALS patients, sometimes hypothetical diagnosis of ALS by doctors would occur frequently. Hence, using un-hospitalized ALS cases from EMRs would run a risk for constructing a wrong prediction model. We insist to use those who were hospitalized due to ALS disease as our positive dataset and their corresponding comorbidity records within previous two years are considered as true effective comorbidities.
-> I understand that restriction to hospitalised cases may make sense if the intention is to construct a prediction model that is as accurate as possible, but it would strenghen the manusctipt if the authors could discuss this and any limitations in the use of the model due to that restriction. If the prediction model is based on a highly selected patient population and then applied to the broader patient population, could the model accuracy be overestimated as it was not accounting for the diversity of patients in the EMRs where it is supposed to be implemented for e.g. screening or early diagnosis in the end? Or will the prediction model work better on ALS patients with certain comorbidities, on the more severe cases, or only on hospitalised patients? Would the prediction model perform worse in detecting early ALS if the model was trained on the broader patient population (all adults including all comorbidities)? It would be interesting for the reader to understand how the authors as experts in this field view that.
Response: Thanks for reviewer’s comments. Thank you for understanding the importance of data cleaning procedures for data analytics. We have clearly defined the problem in our research regarding the ALS patients who had an average of a two-year interval for disease symptoms prior to hospitalization. We just tried to design a prediction model based on our proposed WJI indicator and to validate the prediction accuracy through the real EMR records. It is really not necessary to design a model by analyzing all comorbidities without an interval limitation. The definition of the problem and default time interval were discussed with ALS doctors. The most important thing is that we are not allowed to require any additional EMRs from our governmental database at this stage. We need a new project(budget) and an authorized IRB approval to perform any new simulation.
- LL322 I still do not understand the rationale and impact on the model of capping the control group at ages 49-51, that appears to be a very subjective and selective way and alternative ways should be considered, e.g. matching patients on age and gender like in traditional epidemiology. This has to be explained and I would strongly suggest to apply age and gender matching instead.
Response: Thanks for reviewer’s comments. We have described in the manuscript that the average onset age of the ALS subjects in Taiwan is 51 years old. We would like to know the most popular symptoms of ALS patients (at middle age) instead of other chronic diseases or genetic diseases. Hence, we decided to matching patients on gender only in this study. For example, we have a very young age ALS patient (4 years old, it might occur due to genetic heritage conditions. If we matched with a healthy kid of the same age, and his/her previous EMRs would not gain any valuable information. However, the young ALS patient still possessed similar progressive ALS symptoms as those ALS patients at middle age. Hence, we took subjects in control group at age 49-51.
-> I am concerned about this approach. That would mean that you disregards all other diagnosis-disease associations and create a prediction model that is only based on the associations among patients 49-51? Would that not overestimate the model accuracy and limit its application? I would like to see what the results would be based on a broader age group of eg all adults and at least a discussion about any implications of this in line with the previous comment.
Response: We have defined the problems, designed the prediction models, and validated our initial assumptions in the manuscript. All other additional data request is not possible at this stage. Perhaps the reviewer didn’t understand that all national EMR data should be applied for research and all research projects were strictly evaluated through IRB meetings. We need a new project, funding, and approved IRB document to collect the related EMRs. There is no easy way to perform any new simulation now.
- LL333 Again, the question is if you based the decisions on precision or strength of association? This is inconsistent in the text.
Response: Thanks for reviewer’s comments. Our manuscript is based on strength association through odds ratio analysis. We should avoid using statistically significant in the manuscript.
->To avoid this issue, I would strongly recommend to search the document again, as it appears that the word is still used (e.g. LL158, LL378, LL388, LL511).
Response: Thanks again. It was modified.
NEW COMMENTS
LL119. What is meant by "...verify the effectiveness of the novel proposed indicator" in the aims? I cannot see that there was any test of its performance that would verify its effectiveness. For that, I would expect to see results where this so called indicator was applied an EMR system for screening ALS patients or early diagnosis. LL220 Can you really say that the evaluation is "precise"? On what grounds? Or do you mean "more precise" than the traditional JI? LL221 Did you estimate incidence rates in this study? LL486 Would recommend to describe it as "...using EMRs as the source data to compare comorbidity patterns..." LL488 Would recommend to revise this sentance as it may appear unclear to the reader (...mimicking diseases..."). LL491 Use precise instead of precision.
Response: The results claim that a patient doesn’t need to perform any other invasive examining experiments (traditional ways) to predict the ALS disease compared to using previous two-year EMRs only in the proposed method. It is relatively effective in this sense. The prediction accuracies are the results by using the proposed WJI indicators. We have modified from “precise” to “better” in LL220 for comparison with the other two approaches. We didn’t estimate incidence rates in this study. All the information can be observed from the provided references. “mimicking diseases” was commonly used by ALS societies. Please refer to the references and ALS society web pages(https://alstreatment.com/diseases-that-can-mimic-als/). We have changed the “precision” to “precise” in LL491, Thanks for the suggestions.
Please also find the attachment.

Reviewer 2 Report
In the current revision, the readability of the paper has improved in many places, but in even more places it still has not.
The revised section 2.3 still does not state what input data the classifiers are trained on. A related problem, further impeding understanding, is that the authors repeatedly write "data training", while the common wording would be that some adjustable model is trained on some data. One can only guess that probably, when presented to the classifier, a patient is described by a vector in which comorbidities that the patient has are encoded by Jaccard indices, or weighed Jaccard indices, or scoring weights; and comorbidities the patient does not have are encoded by zeros. However, this guess of mine may be wrong, because in line 187 the authors state that some WJI similarity measurement is compared to some "trained thresholding value".
Furthermore, some normalization procedure is mentioned in line 191 without any explanation.
Generally, many explanations in the methodological part of the paper, in spite of their length, only complicate the subject. For instance, in line 141 the authors have now added a clarification for the notion of minimum support threshold. However, that clarification does not resolve the original issue: it is still not quite clear what actually happens to the comorbidity feature set. If the actual meaning is something as simple as "comorbidities which were present in less than a certain percentage of subjects within the experimental group were discarded from the feature set", it would better have been said in such a simple way. That way, the question about minimum support threshold would not have arisen in the first place.
The explanation of weighed Jaccard indices is done for abstract groups A and B (I apologize for having mistakenly called them groups 1 and 2 in my first review), while it would be better to use the actual group names, EG and CG, instead. Also, in equation (2) the numerator and the denominator are the same, which must be a typo.
In the accuracy metrics, one might recommend the more self-describing names like "accuracy on the testing data" and "accuracy on the training data" instead of "accuracy" and "model score" respectively.
In the accuracy tables, deviation ranges over cross-validation folds are not shown. If there is no more room left for them in the tables, the accuracy values should at least be rounded to their least significant digits.
The structuring of the paper has indeed become better.
Overall, the work done by the authors may well be a contribution to the field, but its presentation still needs revision.
Author Response
We thank the reviewer’s comments. We answered the questions in brown.
In the current revision, the readability of the paper has improved in many places, but in even more places it still has not.
The revised section 2.3 still does not state what input data the classifiers are trained on. A related problem, further impeding understanding, is that the authors repeatedly write "data training", while the common wording would be that some adjustable model is trained on some data. One can only guess that probably, when presented to the classifier, a patient is described by a vector in which comorbidities that the patient has are encoded by Jaccard indices, or weighed Jaccard indices, or scoring weights; and comorbidities the patient does not have are encoded by zeros. However, this guess of mine may be wrong, because in line 187 the authors state that some WJI similarity measurement is compared to some "trained thresholding value".
Response: Thanks reviewer’s comments. We have rewrite the paragraph for better description according to reviewer’s suggestions.
“…The proposed diagnostic model is mainly based on the similarity measurement of comorbidity patterns between testing subjects and known ALS patients. From experimental and control data groups, we initially calculated the odds ratios for each comorbidity (within a two-year interval) for measuring the association between comorbidities and ALS disease. Based on odds ratio analysis, comorbidities with strong associations were selected and constructed as an important comorbidity feature set, and the proposed WJI similarity measurements were further calculated for experimental group vs. the comorbidity feature set and control group vs. the comorbidity feature set respectively. A binary outcome prediction model for supporting clinical decision making was trained on the calculated WJI indicators of experimental (ALS) and control (non-ALS) groups versus ALS associated comorbidity feature set. Once the WJI indicators of experimental and control data groups were trained and constructed as a prediction model, only the selected associated comorbidities of testing subjects were considered and evaluated according to the WJI similarity measurement and compared to the previously trained thresholding value. The ALS prediction module was constructed and described as the followings. Obtaining the comorbidities for each subject in the experimental or control groups, and it’s corresponding WJI similarity was calculated with respect to the defined associated comorbidity feature set. Subsequently, a normalization procedure was performed for similarity measurement and for training and construction of the probability prediction model….”
Furthermore, some normalization procedure is mentioned in line 191 without any explanation.
Responses: Thank for reviewer’s reminding. Since we only have the WJI indicators for the mode construction, we should use standardization instead of normalization. The contexts were modified as the following: (LL191)
“…Subsequently, a Z-Score standardization procedure was performed for similarity measurement and for training and construction of the probability prediction model….”
Generally, many explanations in the methodological part of the paper, in spite of their length, only complicate the subject. For instance, in line 141 the authors have now added a clarification for the notion of minimum support threshold. However, that clarification does not resolve the original issue: it is still not quite clear what actually happens to the comorbidity feature set. If the actual meaning is something as simple as "comorbidities which were present in less than a certain percentage of subjects within the experimental group were discarded from the feature set", it would better have been said in such a simple way. That way, the question about minimum support threshold would not have arisen in the first place.
Responses: Thank for reviewer’s great suggestions. We have re-wrote and added the phrase in the paragraph.
“…A minimum support threshold value represents the minimum number of subjects or the minimum percentage of total subjects within the experimental group. In other words, comorbidities which were present in less than a certain percentage of subjects within the experimental group were discarded from the feature set in this study.…”
The explanation of weighed Jaccard indices is done for abstract groups A and B (I apologize for having mistakenly called them groups 1 and 2 in my first review), while it would be better to use the actual group names, EG and CG, instead. Also, in equation (2) the numerator and the denominator are the same, which must be a typo.
Responses: Thanks for reviewer’s comments. It is not a typo, we used simple notation to represent the non-union comorbidity. We have re-wrote the equation for a better representation.
(Equation, please find in attachment)
In the accuracy metrics, one might recommend the more self-describing names like "accuracy on the testing data" and "accuracy on the training data" instead of "accuracy" and "model score" respectively.
Response: Thanks for the comments, the description were described in the Section of 2.7. Model Evaluation Index.
“…Accuracy was the accuracy rate of prediction based on the testing data; it is defined as (TP+TN)/(TP+FP+TN+FN); the model score is the accuracy rate of prediction obtained from the training data itself….”
In the accuracy tables, deviation ranges over cross-validation folds are not shown. If there is no more room left for them in the tables, the accuracy values should at least be rounded to their least significant digits.
Response: Thanks for the comments. The research results of a k-fold cross-validation are frequently summarized with the mean of the model scores only. I would like to apologized that I didn’t have the data of variances of the scores for all different ML approaches. It is a dilemma for me that the graduated student who serves in the military and is not able to be contacted now. So the simulations are not able to be run in a short time by other students. I only have parts of the data as the following figures. So, if it is not urgently required data, we hope we don’t have to show the variations of the 5-fold cross-validation for each approach on the manuscript. Here are the examples of one of ML approaches to demonstrate what I have now. We really appreciate your understanding.
(Figure, please find attachment).
The structuring of the paper has indeed become better.
Overall, the work done by the authors may well be a contribution to the field, but its presentation still needs revision.
Response: We do appreciate the reviewer’s comments and guidance to make the manuscript greatly improved.
Please also find the attachment.
